# $V_1$: Unifying Generation and Self-Verification for Parallel Reasoners

**Harman Singh**[*1] **Xiuyu Li**[*1] **Kusha Sareen**[2] **Monishwaran Maheswaran**[1] **Sijun Tan**[1] **Xiaoxia Wu**[3] **Junxiong Wang**[3] **Alpay Ariyak**[3] **Qingyang Wu**[3] **Samir Khaki**[1] **Rishabh Tiwari**[1] **Long Lian**[1] **Yucheng Lu**[3] **Boyi Li**[14] **Alane Suhr**[1] **Ben Athiwaratkun**[3] **Kurt Keutzer**[1]

## Abstract

Test-time scaling for complex reasoning tasks shows that leveraging inference-time compute, by methods such as independently sampling and aggregating multiple solutions, results in significantly better task outcomes. However, a critical bottleneck is *verification*: sampling is only effective if correct solutions can be reliably identified among candidates. While existing approaches typically evaluate candidates independently via scalar scoring, we demonstrate that models are substantially stronger at **pairwise verification**. Leveraging this insight, we introduce $V_1$, a framework that unifies generation and verification through efficient pairwise ranking. $V_1$ comprises two components: $V_1$**-Infer**, an uncertainty-guided algorithm using a tournament-based ranking that dynamically allocates self-verification compute to candidate pairs whose relative correctness is most uncertain; and $V_1$**-PairRL**, an RL framework that **jointly trains** a single model as both generator and pairwise self-verifier, ensuring the verifier adapts to the generator's evolving distribution. On competitive code generation, SWE and terminal-based agents, math reasoning, and medical QA benchmarks, $V_1$-Infer improves Pass@1 by up to 10% over pointwise verification and outperforms SoTA aggregation-based test-time scaling while being more efficient. Furthermore, $V_1$-PairRL achieves test-time scaling gains of $3.8 - 10$pp over standard RL and pointwise joint training on code, math, and non-verifiable domains, and improves base Pass@1 by up to 8.7pp over standard RL. [Project page & code](#).

---

[*]Equal contribution [1]UC Berkeley [2]Mila [3]Together AI [4]NVIDIA. Correspondence to: Harman Singh <harmans@berkeley.edu>, Xiuyu Li <xiuyu@berkeley.edu>, Kurt Keutzer <keutzer@berkeley.edu>.

*Proceedings of the $43^{rd}$ International Conference on Machine Learning*, Seoul, South Korea. PMLR 306, 2026. Copyright 2026 by the author(s).

## 1. Introduction

Large language models (LLMs) have demonstrated remarkable problem-solving abilities, largely driven by the paradigm of "System 2" thinking (OpenAI, 2024a;b; DeepSeek-AI, 2025) of generating extended chains of thought to reflect, refine, and verify answers at inference time. *Parallel reasoning*, which complements this sequential "deep thinking" by sampling multiple independent chains of thought to explore diverse solution paths, has emerged as a powerful technique for test-time scaling (Wang et al., 2023; Cobbe et al., 2021; Pan et al., 2025; Lian et al., 2025; Snell et al., 2024). In this setup, parallel sampling of multiple chains-of-thought is followed by an aggregation step to select the final answer. The simplest form of aggregation is to select the most common solution among the set of candidates (majority voting). While this suffices for domains like math where answers are objective and easily verifiable (Wang et al., 2023), this cannot be used for more general domains which do not admit objective ground truth answers. Instead, the ability to accurately *self-verify* solutions can support an aggregation method that selects the correct solution from the candidate set, as long as it exists in the set, even if it isn't the most common solution. Thus, taking full advantage of parallel reasoning fundamentally hinges on *accurate verification*: sampling $N$ solutions is useful if the model can reliably identify the correct one.

Our experiments identify a critical bottleneck in existing approaches that use models' intrinsic verification capabilities to take advantage of inference-time compute: without a globally comparable scale of solution quality, existing models are not calibrated to evaluate candidate solutions independent of one another. In addition, existing work suggests that in such settings, models used as verifiers are biased towards positively evaluating their own samples, even if those samples are incorrect (Lu et al., 2025). We find that, instead, self-verifying between candidate solutions via pairwise comparison leads to more robust and accurate outcomes. We explore two core questions. First, *how can we enable LLMs to more accurately self-verify candidate solutions obtained via parallel reasoning, by leveraging pairwise candidate self-verification?* Second, given that self-verification is typically applied after a model has already been trained, *can*

*we instead train models to be better at self-verification for parallel reasoning?*

While pairwise ranking has been extensively studied in reward modeling for LLM alignment (Christiano et al., 2017; Ziegler et al., 2019), this approach remains underexplored for self-verification in a parallel reasoning setting. Additionally, while reinforcement learning is commonly used to improve the solution-generation capabilities of LLMs on verifiable domains such as code and math (Shao et al., 2024; DeepSeek-AI, 2025), no existing method effectively utilizes the parallel reasoning chains of LLMs at training time to jointly optimize both the generation and self-verification capabilities, resulting in distribution shifts at inference time. We demonstrate that inducing pairwise self-verification capabilities during RL training is an effective technique to improve test-time scaling performance for parallel reasoners. We leverage our observations to develop $\mathbf{V}_1$, a unified framework that includes a strong inference time scaling algorithm for parallel reasoning, as well as a reinforcement learning framework that induces self-verification capabilities during RL training of LLMs with verifiable rewards.

Our contributions include the following:

**1.** We show that in parallelized reasoning, independent self-verification of candidate solutions suffers from calibration collapse due to lack of comparative reference. Additionally, self-aggregation methods like recursive self-aggregation (RSA; Venkatraman et al., 2025) may lead to diversity collapse where Pass@N monotonically decreases with aggregation steps. This motivates pairwise verification as a principled alternative for self-verification and an orthogonal methodology for test-time scaling without loss of diversity.

**2. We develop $\mathbf{V}_1$-Infer**, an uncertainty-guided pairwise verification algorithm. Rather than scoring solutions in isolation, it pairs candidates by employing a Swiss-system tournament refinement strategy that dynamically allocates verification compute to the most uncertain pairs. This approach provides a significant boost in selection accuracy, effectively improving the performance of the model closer to Pass@N of the original sampled responses. Notably, we find that $\mathbf{V}_1$-Infer outperforms or matches RSA on tasks where aggregation leads to diversity collapse, while requiring significantly fewer verification calls.

**3. We develop $\mathbf{V}_1$-PairRL**, an RL framework that co-trains a single model as both generator and pairwise self-verifier. Unlike prior co-training approaches that rely on pointwise rewards (Sareen et al., 2025; Liu et al., 2025a) or offline data (Venkatraman et al., 2025), $\mathbf{V}_1$-PairRL uses an online, co-evolving objective where generation and pairwise verification improve together. This ensures that as the generator improves, the verifier trains on in-distribution data from the model's current capabilities, thereby leading to stronger

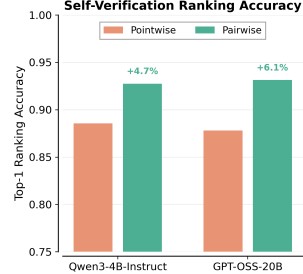
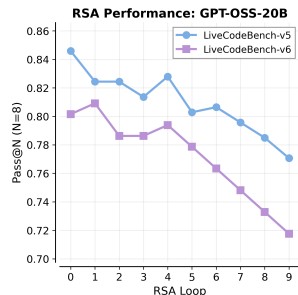

*Figure 1.* (**Left**) Pairwise self-verification (using $\mathbf{V}_1$-Infer, §3) outperforms pointwise self-verification on problems with both correct and incorrect solutions in their parallel generations (GPT-OSS-20B on LiveCodeBench-v6 prompts). (**Right**) Recursive self-aggregation of GPT-OSS-20B on LCB-v5 and v6 shows declining Pass@N (diversity collapse). See §2.

self-verification capabilities at inference time.

We evaluate $\mathbf{V}_1$-Infer on code generation (LiveCodeBench, CodeContests), SWE and terminal-based agents (SWE-Bench, Terminal-Bench), math reasoning (AIME, HMMT), and non-verifiable medical QA (HealthBench) benchmarks. $\mathbf{V}_1$-Infer improves Pass@1 by up to 10% over pointwise verification and matches or exceeds RSA. $\mathbf{V}_1$-PairRL achieves 7–9% test-time scaling gains over pointwise co-training and standard RLVR on code, math, and rubric-based RL on a non-verifiable medical QA domain, and improves base Pass@1 by up to 8.7% over the standard RL. Our results demonstrate that unified training for solution generation and self-verification capability, coupled with our pairwise self-verification technique, enables effective parallel reasoning. A comprehensive discussion of related work is provided in App. A.

## 2. Limitations of Current Self-Verification and Aggregation Approaches

From a test-time scaling perspective without external verifiers, parallel reasoning offers two prevalent mechanisms to generate the final solution: **a) self-selection** (verifying and choosing the best candidate) and **b) self-aggregation** (combining solutions to generate a better one). We analyze the limitations of current approaches in both categories to motivate our pairwise framework [1].

**Pointwise self-verification (llm-as-a-judge) suffers from calibration collapse.** Standard pointwise verification assigns scalar scores to solutions in isolation. This approach is fundamentally limited by the lack of a comparative reference set. Statistically, latent utilities in choice models (e.g., Bradley–Terry) are identifiable only up to monotonic transformations, meaning absolute scores lack a globally comparable scale (Bradley & Terry, 1952). Consequently, point-

---

[1] Majority voting is less general and only applicable to scenarios with objective ground-truth answers, such as math.

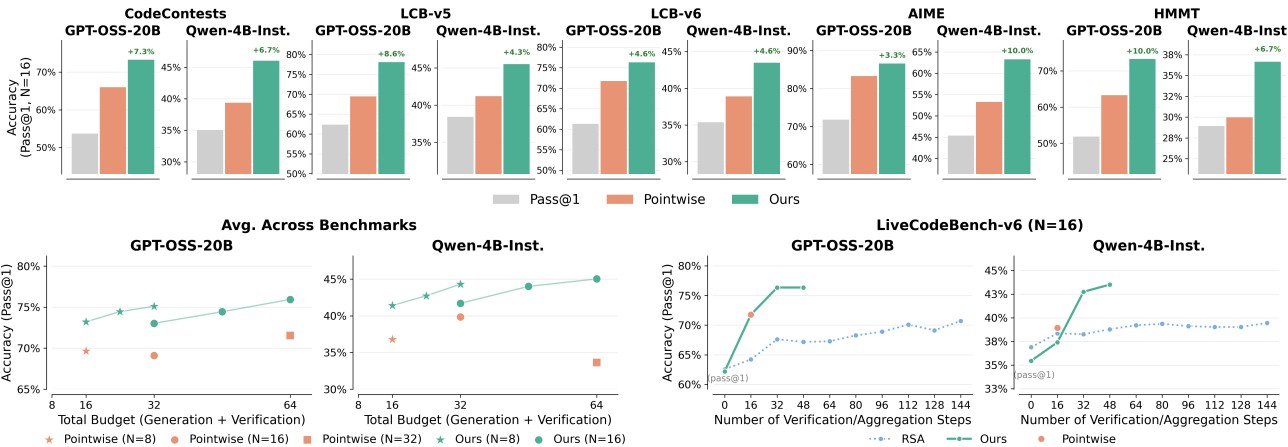

*Figure 2.* **(Top)** Performance after self-verification using $V_1$-Infer compared with pointwise self-verification across benchmarks and models at $N=16$ base generations. Results presented for GPT-OSS-20B and Qwen3-4B-Instruct-2507. **(Bottom Left)** Accuracy vs. total budget (generation + verification calls). Stars, circles, and squares denote $N=8$, $N=16$, and $N=32$ base generations. $V_1$-Infer consistently outperforms pointwise self-verification at equivalent budgets and shows monotonic performance scaling with compute. **(Bottom Right)** Comparison with Recursive Self-Aggregation (RSA) (Venkatraman et al., 2025) on LCB-v6. $V_1$-Infer achieves higher accuracy with fewer self-verification calls. Results for GPT-OSS-120B and Qwen3-4B-Thinking-2507 are in Fig. 11.

wise scores exhibit high variance and poor cross-context calibration, often over-scoring plausible but incorrect solutions. Christiano et al. (2017) noted that for learning from human preferences, relative scores are much easier to provide for humans compared to absolute scores. *Pairwise* judgments simplify the task to a well-posed relative comparison. As shown in Figure 1 (left), this shift to relative ranking yields significantly higher top-1 self-ranking accuracy.

**Self-aggregation leads to reduction in pass@N and diversity collapse.** Self-aggregation-based methods prompt the same LLM to consolidate parallelly generated solutions into one solution. While self-aggregation methods (Venkatraman et al., 2025; Khairi et al., 2025; Li et al., 2025b; Madaan et al., 2025) can help consolidate parallel reasoning chains and improve Pass@1, they may lead to *diversity collapse*. As shown in Figure 1 (right), the Pass@N score, representing the probability that *at least one* correct solution exists in a set of generated N solutions, monotonically decreases as aggregation steps increase for RSA. This indicates that RSA frequently discards or degrades correct outlier solutions during refinement. Specifically, since the refined Pass@1 rarely exceeds the *initial* Pass@N of the raw samples, the value of self-aggregation is unclear compared with a strong self-verifier that can select the best answer and reach near Pass@N performance. Instead of relying on implicit self-verification capabilities of aggregation based method, explicit and accurate self-verification can provide orthogonal improvements to aggregator-based approaches since each aggregation step (that may induce diversity collapse) can benefit from self-verification (that maintains pass@N), which can help select promising candidate solutions to aggregate.

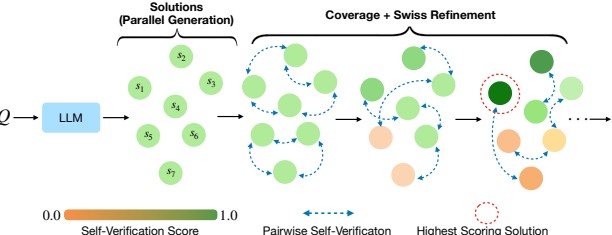

*Figure 3.* Swiss Refinement Overview. Increasing pairwise verifications enables LLMs to better self-verify for selecting the best response among N self-generated solutions. See Section 3.

## 3. Improving the Self-Verification Capability of LLMs using $V_1$-Infer

Pointwise verification is limited in its ability to capture relative nuances and lacks calibration. These limitations, coupled with a desire to maintain solution diversity, motivate us to propose $V_1$-*Infer*, a pairwise self-verification framework designed for parallel reasoning. While we can use pairwise self-verification to score responses by the model, naively pairing solutions may require quadratic number of $C(N, 2)$ self-verification attempts. Our approach, detailed in Algorithm 1, consists of a weighted aggregation mechanism and a two-phase budgeting strategy designed to maximize information gain with each new pair which allows us to efficiently scale test-time self-verification compute while achieving significant improvement in reasoning performance. A high-level overview of our approach is presented in Fig. 3.

**Uncertainty-Guided Score Aggregation.** Standard pairwise voting often treats all "wins" equally, and thus fails to distinguish between a marginal preference (e.g., a 6 vs. 5 rating) and a decisive victory. Instead, we prompt the models to score their solutions in a pairwise setting instead

---

**Algorithm 1** Uncertainty-Guided Pairwise Ranking

---

**Require:** Problem $x$, candidates $\mathcal{S} = \{s_i\}_{i=1}^{N}$, pairwise budget $B$, min-degree $d_{\min}$, Swiss window size $h$, weight floor $\tau$
**Ensure:** Ranking $\pi$
1: **Initialize:** $\forall i$, scores $\mu_i \leftarrow 0.5$ and degrees $d_i \leftarrow 0$
2:  history $\mathcal{H} \leftarrow \emptyset$ $\{(i,j) \in \mathcal{H}$ iff $(s_i, s_j)$ has been compared$\}$
3:  state $\mathcal{T} \leftarrow (\{\mu_i\}_{i=1}^{N}, \{d_i\}_{i=1}^{N}, \mathcal{H})$
4: used $\leftarrow 0$
5: **Phase 1: Topology Coverage**
6: **while** used $< B$ **and** $\exists i : d_i < d_{\min}$ **do**
7:  $\mathcal{P} \leftarrow$ COVERAGEPAIRS$(\mathcal{T}, d_{\min})$
8:  $\mathcal{O} \leftarrow$ PAIRSELFVERIFY$(x, \mathcal{S}, \mathcal{P})$ {parallel LLM judging}
9:  UPDATESTATS$(\mathcal{T}, \mathcal{O}, \tau)$ {update $\mu_i, d_i$; record $\mathcal{H}$}
10:  used $\leftarrow$ used $+|\mathcal{P}|$
11: **end while**
12: **Phase 2: Swiss Refinement**
13: **while** used $< B$ **and** $N > 2$ **do**
14:  $\pi \leftarrow$ RANK$(\{\mu_i\}_{i=1}^{N})$ {descending $\mu$}
15:  $\mathcal{P} \leftarrow$ SWISSPAIRS$(\pi, \mathcal{T}, h)$ {within window $h$; prefer unseen and near-ties}
16:  $\mathcal{O} \leftarrow$ PAIRSELFVERIFY$(x, \mathcal{S}, \mathcal{P})$
17:  UPDATESTATS$(\mathcal{T}, \mathcal{O}, \tau)$
18:  used $\leftarrow$ used $+|\mathcal{P}|$
19: **end while**
20:
21: **return** RANK$(\{\mu_i\}_{i=1}^{N})$

---

of simply providing "correct" or "incorrect" as the output as in standard pairwise voting, which provides us with fine-grained information about the goodness of solutions. To capture this nuance of relative quality, we adopt a weighted aggregation scheme in which the magnitude of the rating difference serves as a proxy for the judge's confidence. Given $N$ candidate solutions $\mathcal{S}$ generated by an LLM, and a comparison budget $B$ (number of self-verification LLM calls), a pairwise comparison between $(s_i, s_j)$ using the same LLM outputs calibrated ratings $(r_i, r_j) \in [1, 10]$ for the two solutions. We define a confidence weight

$$w_{ij} = \max\left(\frac{|r_i - r_j|}{9}, \tau\right),$$

where $\tau > 0$ is a small floor that ensures non-zero weights even for near-ties. Let $v_{ij} \in \{0, 0.5, 1\}$ denote the comparison outcome for $s_i$, corresponding to a win, tie, or loss, respectively. The estimated quality score $\mu_i$ is then computed as the uncertainty-weighted win rate

$$\mu_i = \frac{\sum_{j \in \mathcal{N}(i)} w_{ij} v_{ij}}{\sum_{j \in \mathcal{N}(i)} w_{ij}}, \quad (1)$$

where $\mathcal{N}(i)$ denotes the set of opponents compared against $s_i$. This formulation ensures that high-confidence judgments dominate the global ranking, while ambiguous comparisons contribute minimally to score variance.

**Phase 1: Topology Coverage.** A primary failure mode in low-budget pairwise ranking is *path dependence* in which solutions become "orphaned" or misranked due to insufficient

pairwise comparisons with other solutions. We mitigate this by enforcing a minimum degree constraint to ensure all solutions are pairwise self-verified at least a minimum number of times. We begin with random disjoint pairings to guarantee global connectivity ($d_i \geq 1$), ensuring every solution enters the tournament. Subsequently, we iteratively target under-sampled nodes ($d_i < d_{\min}$) to meet a minimum degree threshold. Rather than selecting random opponents, we pair these low-degree nodes with candidates having the closest current mean score $\mu$. This "anchors" solutions against comparable peers early in the process, preventing initial noise from propagating into the refinement phase.

**Phase 2: Swiss Refinement.** With the topology anchored, the remaining budget focuses on resolving rank ambiguity through an uncertainty-aware Swiss system. In each round, solutions are sorted by their current score $\mu$, and we pair neighbors within a local window to minimize the score gap $|\mu_i - \mu_j|$ among unseen pairs. This strategy is grounded in active learning principles: under Bradley-Terry models, comparisons between items of similar skill (near-ties) yield the highest marginal information gain. By concentrating the judge budget on these ambiguous decision boundaries, $\mathbf{V}_1$-Infer efficiently reduces uncertainty where it matters most, achieving high ranking accuracy even with a sparse comparison graph ($K \ll N$). See Appendix §F for the detailed algorithm and full specification of the helper procedures.

> **Note (Generality Beyond Self-Verification):** While most of our experiments (except on Terminal-Bench) are in a "self-verification" setup, i.e., using the same model as generator and verifier, our approach is highly general and applicable to any LLM-based verification setting with different models.

### 3.1. Experimental Settings

**Models and Benchmarks.** We evaluate four diverse models: GPT-OSS-20B, Qwen3-4B-Instruct, GPT-OSS-120B, and Qwen3-4B-Thinking. For code generation, we use LiveCodeBench-v5 (279 problems between date range of 24.08 to 25.02), LiveCodeBench-v6 (131 problems between 25.02-25.05) (Jain et al., 2024), and CodeContests (165 problems) (Li et al., 2022). For SWE and terminal-based agents, we evaluate $\mathbf{V}_1$-Infer on SWE-bench Lite (Jimenez et al., 2024) (300 instances) using Gemini-2.5-Flash (Comanici et al., 2025) as both the generation and verification model, and on Terminal-Bench (89 tasks) using GPT-5.4 trajectories from the ForgeCode harness with `gpt-5-mini` as the (weaker) verifier (full setup in §3). For math, we use AIME'25 and HMMT'25 (Balunović et al., 2025). For non-verifiable open-domain QA, we evaluate on Health-Bench Main and HealthBench Hard (Arora et al., 2025) (200 prompts per benchmark), sampling $N{=}16$ responses per prompt and grading them with HealthBench's physician-

curated rubrics using GPT-5-nano as the grader.

**Evaluation Protocol and Baselines.** For all methods, we first generate $N$ candidate solutions independently from the model, then apply the verification strategy to select the final answer. We use $N \in \{8, 16\}$ for most experiments, and additionally $N = 32$ for pointwise verification in budget-matched comparisons. For comparison with RSA (Venkatraman et al., 2025) we keep the population size as $N = 16$ with $k = 4$ as the aggregation size (number of solutions to aggregate at a time), with $T = 10$ RSA loops. Details about the parameters and settings for inference sampling and swiss-refinement algorithm are provided in Appendix E. We prompt the models to provide a fine-grained score between 1-10 for each response when performing pairwise verification. Prompts are provided in Appendix K.

**Pointwise (llm-as-a-judge) Baseline.** Pointwise self-verification uses the same 1–10 grading system as our pairwise method. The model evaluates each solution in isolation and the solution with the highest score is selected.

**Verification Budget.** $\mathbf{V}_1$-Infer allows flexible control over the verification budget. We report results for budget multipliers of $1\times$, $2\times$, and $3\times$ the number of initially generated solutions ($N$). For example, with $N = 16$ and budget $2\times$, we perform 32 pairwise comparisons.

## 3.2. Results

Our goal is to measure the efficacy of $\mathbf{V}_1$-Infer compared to standard LLM-as-a-judge (pointwise) verification for parallel reasoners, and analyze how it compares with pointwise and aggregation-based test-time scaling methods in a budget-matched setting. Figure 2 provides an overview of our results, comparing pointwise verification with pairwise verification, budget-matched evaluation, and budgeted comparison with Recursive Self-Aggregation.

**$\mathbf{V}_1$-Infer consistently outperforms pointwise (llm-as-a-judge) self-verification.** On CodeContests, GPT-OSS-20B improves from 66.06% to 73.33% (+7.3%), while Qwen3-4B-Instruct improves from 39.4% to 46.1% (+6.7%). On LiveCodeBench-v5, GPT-OSS-20B gains +8.6% and Qwen3-4B-Instruct gains +4.3%. On HMMT, GPT-OSS-20B gains +10.0% and Qwen3-4B-Instruct gains +6.7%. See Figure 2 (top) for more results and Appendix Figure 11 for results on Qwen-4b-Thinking-2507 and GPT-OSS-120B. These results demonstrate that pairwise comparisons provide more informative signals than independent pointwise scores. We further observe that as verification budget increases, pairwise verification performance improves or stays consistent across most models and benchmarks (see Appendix Figure 11). While the results above are for the same number of base solution generations (N=16), we find similar trends while comparing methods in a compute-matched

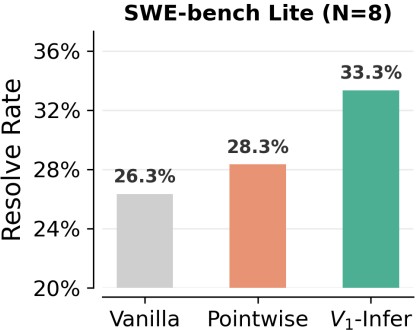

*Figure 4.* Pairwise comparison with vanilla solution generation and pointwise self-verification on SWE-bench Lite (300 instances, $N$=8 candidates, Gemini 2.5 Flash, mini-swe-agent (Yang et al., 2024) pipeline).

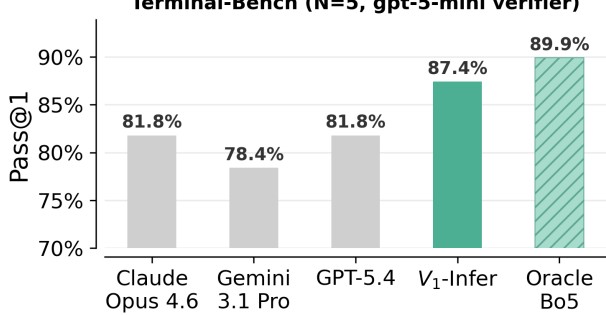

*Figure 5.* Terminal-Bench (89 tasks, ForgeCode harness). $\mathbf{V}_1$-Infer over $N$=5 GPT-5.4 trajectories with `gpt-5-mini` as the (weaker) verifier reaches $87.4\%$ Pass@1, beating all single-model baselines and closing $\sim 70\%$ of the gap to Oracle Bo5.

setting as well, Figure 2 (bottom left), e.g., with a compute budget of 64 model calls, with Qwen3-4B-Instruct-2507, pointwise verification performs approx. 33% while pairwise verification reaches 45% accuracy on average on code gen. benchmarks. Multi-seed (3 seeds) results with mean ± standard deviation are reported in Appendix H.

**$\mathbf{V}_1$-Infer enables improved test-time scaling.** We compare against Recursive Self-Aggregation (RSA) (Venkatraman et al., 2025), a state-of-the-art test-time scaling method that iteratively refines solutions through an evolutionary self-aggregation process. As shown in Figure 2 (bottom right), $\mathbf{V}_1$-Infer achieves higher accuracy with significantly fewer LLM calls. On LiveCodeBench-v6 with N=16, our method reaches 76% Pass@1 with only 48 verification calls, higher compared to the maximum accuracy attained by RSA. This efficiency stems from our uncertainty-guided Swiss refinement, which concentrates comparisons on informative pairs rather than exhaustively aggregating or verifying all solution pairs. See Figures 12, 13, 14 for extended results.

**Verifying Agentic Coding and Terminal-based Agents.** We apply $\mathbf{V}_1$-Infer to two agentic settings. ***SWE-bench Lite*** (Jimenez et al., 2024) (300 GitHub issues): Gemini-2.5-

Flash generates $N{=}8$ candidate patches per instance via the mini-swe-agent (Yang et al., 2024) harness and self-verifies by looking at only the issue description and the patch diffs. $\mathbf{V}_1$-Infer reaches 33.3% resolve vs. 28.3% for pointwise and 26.3% for vanilla first-candidate selection (Fig. 4), showing that head-to-head patch comparison surfaces subtle correctness differences (e.g., root-cause vs. surface-level fixes) that pointwise scoring misses. *Terminal-Bench* (89 tasks): GPT-5.4 with the ForgeCode harness produces $N{=}5$ candidate trajectories (Pass@1 = 81.8%, oracle Bo5 = 89.9%). $\mathbf{V}_1$-Infer at $3N$ verification budget with a comparatively *weaker* verifier (gpt-5-mini) reaches 87.4% (Fig. 5), beating multiple single-model baseline evaluated under the same harness — Claude Opus 4.6, Gemini 3.1 Pro, and GPT-5.4 — and closing ∼70% of the gap from Pass@1 to oracle Bo5. This indicates that pairwise self-verification (i) scales to multi-turn agent trajectories and (ii) remains effective even when the verifier is weaker than the generator. See Appendix L for qualitative SWE-bench examples.

**Verification for non-verifiable open-domain medical QA.** Following the HealthBench setup described above, Table 1 reports $\mathbf{V}_1$-Infer vs. pointwise self-verification across three off-the-shelf models on HealthBench Main and HealthBench Hard. $\mathbf{V}_1$-Infer outperforms pointwise on all 6 model×split combinations, with the largest relative gains (+10% to +27%) on HealthBench Hard, mirroring the difficulty-stratified trend observed on code (Section 3). We observe that RSA (Venkatraman et al., 2025) suffers severe *diversity collapse* on HealthBench Hard with gpt-oss-120b, where Pass@1 drops from 0.303 at loop 0 to 0.035 within three aggregation steps, in line with Sec. 2.

*Table 1.* $\mathbf{V}_1$-Infer vs. pointwise self-verification on non-verifiable medical QA. $N{=}16$, GPT-5-nano grader.

| Benchmark | Model | $\mathbf{V}_1$-Infer | Pointwise | Rel. Gain |
|---|---|---|---|---|
| HB-Hard | gpt-oss-20b | **0.219** | 0.172 | +27% |
| | gpt-oss-120b | **0.356** | 0.324 | +10% |
| | Qwen3-4B-Instruct | **0.170** | 0.135 | +26% |
| HB-Main | gpt-oss-20b | **0.495** | 0.474 | +4% |
| | gpt-oss-120b | **0.586** | 0.571 | +3% |
| | Qwen3-4B-Instruct | **0.420** | 0.391 | +7% |

### 3.3. Analysis and Ablations

**Performance gains across different difficulty levels.** Figure 6 shows how $\mathbf{V}_1$-Infer improves accuracy across different problem difficulty levels. On easy problems, Pass@1 is already near-optimal (99.3%). However, on hard problems where Pass@1 is only 40.2%, pairwise verification with budget 3x achieves 63.9%, a gain of +23.7%. Medium difficulty problems show an improvement of +15.4%. This pattern demonstrates that $\mathbf{V}_1$-Infer is most valuable precisely where it is needed most: on challenging problems where the model generates a diverse set of candidate solutions and accurate selection among them is critical for bridging the gap between Pass@1 and Pass@N.

**$\mathbf{V}_1$-Infer complements aggregation and improves over external verifiers.** Pairwise self-verification provides an explicit fitness signal that, when injected into an evolutionary self-aggregation loop (RSA (Venkatraman et al., 2025), AlphaEvolve (Novikov et al., 2025)), accelerates convergence: as shown in Figure 7, $\mathbf{V}_1$-Infer-guided RSA converges to 93.3% on AIME with lower cumulative latency than vanilla RSA or pointwise-guided RSA. Appendix analyses further show that $\mathbf{V}_1$-Infer (i) beats an uncertainty-blind random-pair baseline (§G.1), (ii) is $2.0-2.7\times$ more token-efficient than RSA at higher accuracy and matches or beats pointwise $N{=}32$ on AIME / LCBv6 (§I), (iii) outperforms external reward / process reward models (AceMath-7B-RM, Skywork-o1-Open-PRM, Qwen2.5-Math-PRM-7B, ThinkPRM-7B) at best-of-$N$ on AIME 2025 (§G.2), and (iv) is less length-biased than pointwise on HealthBench (§G.3). Qualitative examples illustrating pointwise *score saturation* are in §M.

> **Takeaway: Improved Self-Verification and Test-Time Scaling with $\mathbf{V}_1$-Infer**
>
> Pairwise self-verification ($\mathbf{V}_1$-Infer) yields more accurate candidate ranking than pointwise (llm-as-a-judge) scoring and enables improved performance by test-time scaling of verification compute, compared with self-aggregation-based test-time scaling. This holds across code generation, math reasoning, agentic tasks, and non-verifiable domains.

## 4. $\mathbf{V}_1$-PairRL: Improving Self-Verification via Unified RL Training

The previous section demonstrated that pairwise self-verification significantly outperforms pointwise approaches at inference time. In the remainder of the paper we consider a second question: *can we explicitly train models to become stronger self-verifiers?* Current RL paradigms for reasoning focus almost exclusively on optimizing the generation of correct solutions, treating verification as an afterthought or an external process. While recent work has explored co-training generators with verifiers (Sareen et al., 2025; Liu et al., 2025a), these approaches rely on *pointwise* rewards and fail to leverage the parallel responses that techniques like GRPO naturally generate during training. Additionally, pointwise verification is uncalibrated, which may make optimization hard. Other methods train for aggregation awareness (Venkatraman et al., 2025; Zhao et al., 2025b) but use offline data, limiting the model's ability to adapt to its own evolving generation distribution during RLVR.

We propose $\mathbf{V}_1$-PairRL, a unified RL framework that trains a single LLM to be both a strong reasoner and an accurate *pairwise* self-verifier. The key insight is that generation and

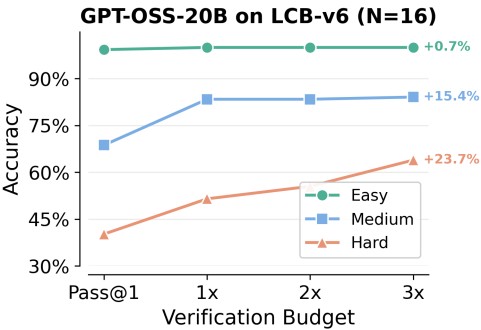

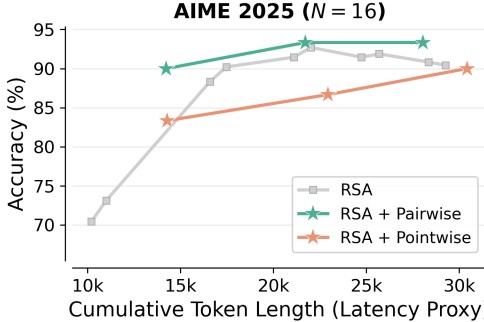

*Figure 6.* Accuracy improvement with increasing verification budget across problem difficulty levels (GPT-OSS-20B on LCB-v6, $N$=16); $\mathbf{V}_1$-Infer provides the largest gains on hard problems.

*Figure 7.* Combining self-verification with RSA on AIME 2025; pairwise self-verification reaches 90% accuracy early and converges to 93.3% with lower cumulative latency than vanilla RSA.

verification should *co-evolve*: as the generator improves, the distribution of responses changes, and the verifier must learn score increasingly high-quality solutions. This online, co-evolving setup ensures verification training data is always in-distribution for the model's current capabilities. However, unified training introduces unique challenges: naive implementations suffer from reward hacking, where the generator and verifier collude to maximize reward without improving actual capabilities. We address these challenges through careful reward design and pairing strategies.

### 4.1. Preliminaries: RL for LLMs

The goal of RL for LLMs is to maximize expected reward $r$ of an LLM policy $\pi_\theta$ over a distribution of prompts $P(Q)$. This is commonly done with policy gradient methods. Specifically, Group-Relative Policy Optimization (GRPO) (DeepSeek-AI, 2025) and its variants (Yu et al., 2025) have shown significant promise and stable optimization dynamics. In each episode, we sample prompts $q \sim P(Q)$ and a group of $G$ rollouts per prompt $\{o_i\} \sim \pi_{\text{old}}(.|q)$. We maximize $J_{\text{Gen}}(\theta)$:

$$\underset{q,\{o_i\}}{\mathbb{E}} \left[ \frac{1}{\sum_{i=1}^{G} |o_i|} \sum_{i=1,t=1}^{G,|o_i|} \min\left( \rho_{i,t} A_i, \text{clip}(\rho_{i,t}, 1-\epsilon, 1+\epsilon) A_i \right) \right]$$
(2)

where $A_i = r_i - \text{mean}(\mathbf{r})$ is the advantage computed over the group and $\rho_{i,t} = \pi_\theta(o_{i,t}|q, o_{i,<t})/\pi_{\text{old}}(o_{i,t}|q, o_{i,<t})$ is the importance sampling ratio.

### 4.2. Co-evolving Solver-Verifier Training

Our training objective combines generation and pairwise verification in a unified RL formulation:

$$J(\theta) = J_{\text{Gen}}(\theta) + \lambda J_{\text{PairVerif}}(\theta)$$
(3)

$J_{\text{Gen}}$ optimizes the likelihood of correct reasoning paths (using GRPO) and $J_{\text{PairVerif}}$ optimizes the accuracy of pairwise ranking judgments. Both objectives use the *same* rollouts: during each training step, the model generates $G$ solutions

per problem, which are used both to compute generation rewards and to form solution pairs for verification training. This ensures the verifier always trains on in-distribution data from the current policy. For verification training, we do one LLM call for each input pair of solutions. Figure 8 shows an overview of $\mathbf{V}_1$-PairRL. Our framework is general, and we show experiments on RL for code and math reasoning, as well as rubric-based RL in non-verifiable domains.

**Rewards.** For the solution generator, we use a standard binary correctness reward $r_{\text{gen}} \in \{0, 1\}$ based on passing all ground truth test cases. Reward is set to 0 if response formatting is incorrect, i.e., the model does not output the response within thinking tags as mentioned in the prompt (§J). For the pairwise self-verification objective, the model compares two solutions $(s_A, s_B)$ and outputs a confidence score between 1 and 10, which is normalized to $v_i \in [0, 1]$ for each solution. We reward the verifier based on how well its scores align with ground truth correctness $y_i \in \{0, 1\}$:

$$r_{\text{verif}} = \frac{1}{2} \sum_{i \in \{A, B\}} \mathbb{I}(|v_i - y_i| \le 0.2) \cdot (1 - |v_i - y_i|) \quad (4)$$

The indicator function $\mathbb{I}(|v_i - y_i| \le 0.2)$ implements a *sparsity threshold*: the verifier receives reward only when its score is within 0.2 of the ground truth (i.e., scoring a correct solution $\ge 0.8$ or an incorrect solution $\le 0.2$). This design choice is critical for preventing reward hacking, which we discuss below. Also see Section §C for implementation details of rubric-based RLVR.

**Mitigating Reward Hacking.** Unified training of generators and verifiers is prone to specific collapse modes. We address two critical forms of reward hacking:

1. *The Safe Bet Collapse:* Without the **sparsity threshold**, the verifier learns to output a safe, middle-ground score (e.g., $v_i = 0.5$) for every solution. This minimizes the risk of being "very wrong" but yields a meaningless discriminator. Sparsity threshold forces the model to commit to confident judgments: only scores near 0 or 1 receive positive reward.

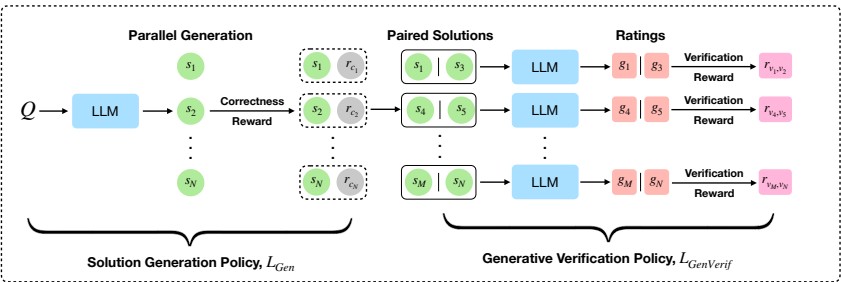

*Figure 8.* **V₁-PairRL: Unified RL training for co-evolving generation and pairwise verification.** A single LLM is trained with two objectives: $J_{\text{Gen}}$ optimizes solution generation using correctness rewards, while $J_{\text{PairVerif}}$ optimizes pairwise verification accuracy. The generator produces $G$ solutions per problem, which are evaluated for correctness and paired for verification training. The verifier evaluates solutions in a paired setting by providing correctness scores for both solutions. Since we are in a verifiable setting and know from ground truth (such as test case execution during code generation) which of the generator solutions are correct, we can calculate correctness rewards for the verifier as well. See Section 4 for more details on how optimization proceeds and rewards are calculated in this framework.

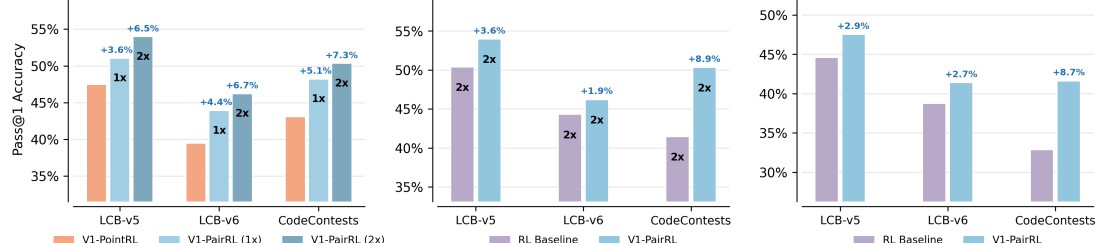

*Figure 9.* **V₁-PairRL training results on Code-RL (with N=16 at inference). (left)** Test-time scaling: **V₁**-PairRL outperforms **V₁**-PointRL and improves with increased verification budget. **(middle)** With **V₁**-Infer as the test-time scaling algorithm (at 2x budget): **V₁**-PairRL outperforms RL baseline even when both use pairwise verification. **(right)** Base Pass@1 accuracy without any test-time scaling: Co-training with pairwise verification improves generation quality over the RL baseline, even without test time scaling.

2. *The Empty Solution Loop:* If the verifier is trained on pairs of two incorrect solutions, the generator may collapse into producing empty or trivially incorrect outputs. The verifier easily identifies these as incorrect (scoring them near 0), receiving high reward. This creates a situation where the generator degrades to maximize the verifier's ease of judgment. To prevent this, we enforce a strict pairing strategy: we only trigger verification training when we can form pairs containing at least one correct solution (Correct-Incorrect or Correct-Correct pairs).

### 4.3. Experimental Settings

We perform experiments with **V₁**-PairRL on RLVR for code generation following the DeepCoder recipe (Luo et al., 2025), math reasoning following (Sareen et al., 2025) and rubric-based RL following (Gunjal et al., 2025).

**Pairwise Verification Training.** For $J_{\text{PairVerif}}$, we group multiple verification prompts (problem + solution pairs from the solver) rather than generating multiple rollouts for a single prompt. This strategy allows us to leverage information from all pairwise comparisons without increasing the total rollout budget. The advantage for verification rollouts reduces to a REINFORCE-style estimator with a mean baseline calculated across prompts.

**Baselines.** We compare our approach against two primary

baselines: (1) a standard **RL baseline** trained solely for generation without a verification objective, and (2) **V₁-PointRL**, a model jointly trained with pointwise verification rewards under the same setup. Training details for **V₁**-PointRL are in App. B. See App. C for additional details about baselines and methodology used for implementing math-RL as well as the rubric-RL setting where the generation reward comes from an external capable judge.

**Models, Datasets, and Training Setup.** We instantiate **V₁**-PairRL on three RL settings: **(i) code RL** (`Qwen3-4B-Instruct-2507` on DeepCoder with DAPO + Dr. GRPO for 150 steps), **(ii) math RL** (`Qwen3-4B-Base` on Polaris-53K with GRPO for ∼300 steps, following Sareen et al. (2025)), and **(iii) rubric-based RL** (`Qwen3-4B-Instruct` on the non-verifiable RaR-Medicine (Gunjal et al., 2025) with GRPO + LLM-judge rubric rewards for 100 steps). Co-evolving runs split rollouts between solver and verifier; **V₁**-PointRL is identical except the pairwise loss is replaced by pointwise scoring. We evaluate the trained models on LiveCodeBench V5/V6 + CodeContests (code), MATH-Hard + AIME 2025 (math, N=16), and HealthBench Main + a held-out RaR-Medicine split (rubric, GPT-5-nano grader). Per-setting hyperparameters, rollout budgets, prompts, and evaluation/inference hyperparameters are in App. C and App. E.1.

## 4.4. Results

**$\mathbf{V}_1$-PairRL Offers Superior Test-Time Scaling Capabilities.** To evaluate the test-time scaling benefits of models co-trained for pairwise verification, we compare $\mathbf{V}_1$-PairRL against $\mathbf{V}_1$-PointRL, our pointwise verification baseline trained with the same co-evolving setup (as described in Appendix B). As shown in Figure 9(left), on Code-RL experiments, pairwise self-verification consistently outperforms pointwise across all benchmarks at N=16. On LiveCodeBench-v5, $\mathbf{V}_1$-PairRL achieves 53.9% (2x budget) compared to 47.4% for $\mathbf{V}_1$-PointRL (+6.5%). Similar improvements are observed on LiveCodeBench-v6 (+6.8%) and CodeContests (+7.3%). Accuracy scales monotonically with verification budget, indicating that the model learns to leverage additional pairwise comparisons. The same advantage holds beyond code. On math RL, $\mathbf{V}_1$-PairRL improves over $\mathbf{V}_1$-PointRL by +5.7pp on MATH-Hard ($62.5 \rightarrow 68.2$) and doubles AIME 2025 accuracy from 10.0 to 20.0 (Table 2), while also raising learned verification accuracy from 63.9 to 71.5 on MATH-Hard. On rubric-based RL in a medical QA setting, $\mathbf{V}_1$-PairRL improves over $\mathbf{V}_1$-PointRL by +3.8pp on HealthBench (0.275 vs. 0.237) and +2.5pp on the held-out RaR-Medicine test split (Table 3).

*Table 2.* $\mathbf{V}_1$-PairRL vs. $\mathbf{V}_1$-PointRL on math RL. Both rows report LLM-best-of-$N$ Pass@1 after applying their respective self-verifiers. $\mathbf{V}_1$-PairRL improves both LLM-based selection accuracy and learned verification accuracy over its pointwise counterpart.

| Method | MATH-Hard | AIME 2025 | Verif. Acc. |
|---|---|---|---|
| $\mathbf{V}_1$-PointRL | 62.5 | 10.0 | 63.9 |
| **$\mathbf{V}_1$-PairRL (Ours)** | **68.2** | **20.0** | **71.5** |

*Table 3.* $\mathbf{V}_1$-PairRL vs. $\mathbf{V}_1$-PointRL on rubric-based RL in a non-verifiable medical QA setting. Here, $N=16$.

| Benchmark | Method | Pass@1 w/ self-verif. |
|---|---|---|
| HealthBench | $\mathbf{V}_1$-PointRL | 0.237 |
| | **$\mathbf{V}_1$-PairRL (Ours)** | **0.275** |
| RaR-Medicine Test | $\mathbf{V}_1$-PointRL | 0.481 |
| | **$\mathbf{V}_1$-PairRL (Ours)** | **0.506** |

**$\mathbf{V}_1$-PairRL outperforms RL Baseline with $\mathbf{V}_1$-Infer as the inference algorithm.** We apply $\mathbf{V}_1$-Infer at inference time to both $\mathbf{V}_1$-PairRL and the standard RL baseline using identical 2x verification budgets. As shown in Figure 9 (middle), $\mathbf{V}_1$-PairRL consistently outperforms the RL baseline even when both leverage pairwise verification: +3.6% on LiveCodeBench-v5, +1.9% on LiveCodeBench-v6, and +8.9% on CodeContests. This shows that co-training with pairwise verification enhances overall performance in an equivalent test-time scaling setup. The same pattern holds on math RL: $\mathbf{V}_1$-PairRL's best-of-$N$ Pass@1 of 20.0 on AIME 2025 beats the no-verifier RL baseline whose base-generation Pass@1 is 15.6, while $\mathbf{V}_1$-PointRL actually *hurts*

relative to that baseline, reaching only 10.0. On rubric-based RL, $\mathbf{V}_1$-PairRL's Pass@1 of 0.275 on HealthBench exceeds a baseline GRPO model trained with the same rubric rewards but without any self-verification co-training (0.256).

**$\mathbf{V}_1$-PairRL improves RL Baseline.** Co-training for pairwise self-verification yields substantial improvements in generation quality. As shown in Figure 9(right), $\mathbf{V}_1$-PairRL achieves consistent Pass@1 improvements over the RL baseline across all three code generation benchmarks at N=16: +2.9% on LiveCodeBench-v5, +2.7% on LiveCodeBench-v6, and +8.7% on CodeContests. The gains indicate that jointly optimising generation and pairwise verification creates a learning signal that improves the model's underlying reasoning, not just its verification accuracy. Going beyond the pointwise-co-training result of Sareen et al. (2025), $\mathbf{V}_1$-PairRL also beats $\mathbf{V}_1$-PointRL in base-generation quality, most clearly on CodeContests (+6%; App. Fig. 15). The same base-generation improvement holds on math RL: on MATH-Hard, $\mathbf{V}_1$-PairRL's base Pass@1 reaches 62.5, versus 58.7 for $\mathbf{V}_1$-PointRL and 56.5 for the no-verifier baseline GRPO. The pairwise judge reward during training rises from 0.19 to 0.59, confirming that pairwise self-verification capability is itself *trainable* from a weak start.

**Co-evolving training is necessary.** An ablation against a non-co-evolving variant (verifier trained on offline base-model solutions) shows that a co-evolving setup consistently outperforms the offline alternative across all code benchmarks; see Appendix D.

> **Takeaway: Unifying Generation and Self-Verification**
>
> Unified RL training that jointly optimizes generation and pairwise verification ($\mathbf{V}_1$-PairRL) produces stronger reasoning models and improves test-time scaling compared with generation-only baselines or models co-trained with pointwise verification.

## 5. Conclusion

We presented $\mathbf{V}_1$, a unified framework for advancing self-verification in parallel reasoning, grounded in the insight that pairwise comparison constitutes a fundamentally more robust primitive for verification than absolute scoring. We introduced $\mathbf{V}_1$-Infer, which employs tournament-based refinement to dynamically allocate compute toward ambiguous pairs, significantly outperforming pointwise verification and prior aggregation techniques. We further improved these results by complementing $\mathbf{V}_1$ with a post-training approach, $\mathbf{V}_1$-PairRL, which demonstrated that jointly training a single model for both generation and pairwise self-verification unlocks superior test-time scaling compared to standard RL and pointwise baselines. By unifying generation with pairwise verification, $\mathbf{V}_1$ provides a robust framework for both effective RL training and scalable parallel reasoning.

## Acknowledgements

We acknowledge the gracious support from the Furiosa AI, Apple, NVIDIA, Macronix, Mozilla team, Open Philanthropy / Coefficient Giving, and Amazon Research. Furthermore, we appreciate the support from Google Cloud, the Google TRC team Prof. David Patterson, along with support from Google Gemini team, and Divy Thakkar. We also acknowledge support by the Director, Office of Science, Office of Advanced Scientific Computing Research, of the U.S. Department of Energy under Contract No. DE-AC02-05CH11231. We thank Aditi Khandelwal, Pragya Srivastava, and Divy Thakkar for their feedback and suggestions on this work. We thank Modal for providing compute credits through the Modal for Academics program. Our conclusions do not necessarily reflect the position or the policy of our sponsors, and no official endorsement should be inferred.

## Impact Statement

This paper presents work whose goal is to advance the field of Machine Learning. There are many potential societal consequences of our work, none which we feel must be specifically highlighted here.

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

# A. Extended Related Work

**Parallel reasoning and test-time scaling.** Test-time scaling approaches fall into two paradigms: sequential methods such as long chain-of-thought that iteratively refine a single solution (Madaan et al., 2023), and parallel methods that generate multiple reasoning paths simultaneously (Cobbe et al., 2021; Setlur et al., 2025; Pan et al., 2025; Zhao et al., 2025a; Singhi et al., 2025; Lian et al., 2025). Parallel scaling helps explore diverse solution paths simultaneously; however, it requires effective mechanisms to combine various solutions. For math, majority voting is effective due to objective answers (Wang et al., 2023; Snell et al., 2024; Wu et al., 2025), but this approach is domain-specific and unavailable for domains such as code generation and scientific discovery, where answers are not directly comparable to each other via exact matching. For code generation, prior work (Li et al., 2025a) has explored test-time scaling but requires executable test cases in the evaluation data or generated by larger models like GPT-4 for clustering and selection, while $\mu$Code (Jain et al., 2025) leverages external, multi-turn execution feedback, both of which are specific to code generation. We focus on *self-verification* for parallel reasoners, where a model judges its own generations without external verified or feedback.

**Self-verification and self-aggregation** Early work by Weng et al. (2023) demonstrated that LLMs can verify their own chain-of-thought reasoning, improving accuracy on arithmetic and commonsense tasks. Related to this, there have been multiple studies analysing the self-refinement capabilities and limitations of LLMs (Madaan et al., 2023; Stechly et al., 2025; Huang et al., 2024), however, they focus on a sequential reasoning setting, while our focus is on explicit self-verification in parallel reasoning. Recently Lu et al. (2025) show that LLMs exhibit a bias toward accepting incorrect solutions during pointwise self-verification. On the other hand, self-aggregation methods like RSA (Venkatraman et al., 2025) and parallel-distill-refine (Madaan et al., 2025) combine solutions generated by the same model, but suffer from diversity collapse (Section 3), resulting in correct solutions being discarded during inference. Such aggregation-based approaches are orthogonal to self-verification and can be combined with better self-verification for more better performance and efficient test-time scaling. Our work addresses limitations of the above prior work by studying pairwise self-verification in a parallel reasoning setup, showing it significantly outperforms pointwise self-verification while preserving solution diversity.

**Generative verifiers and reward models.** Reward models have been central to LLM alignment and RLHF (Christiano et al., 2017; Ziegler et al., 2019; Stiennon et al., 2020), and have emerged as a natural mechanism for scoring and selecting LLM generations (Cobbe et al., 2021; Zheng et al., 2023). Recent work demonstrates that generative reward models, which produce chain-of-thought reasoning before scoring, substantially outperform discriminative approaches (Mahan et al., 2024; Zhang et al., 2025a; Shi & Jin, 2025; Saha et al., 2025), with RL-trained verifiers showing improved judgment performance (Liu et al., 2025c; Whitehouse et al., 2025). Recognizing that pairwise comparison is often easier than absolute scoring, several works have explored pairwise reward models: PairRM (Jiang et al., 2023) trains a discriminative pairwise ranker for ensembling LLM outputs, while GenSelect (Toshniwal et al., 2025) and concurrent work by Mahdavi et al. (2025) scale generative verification for mathematical reasoning. Related to this, Zhao et al. (2025b) propose AggLM, which trains an aggregator via RL to synthesize correct answers for improving majority voting. However, these approaches use *separate* verifiers or aggregator models. Our work differs in two key aspects: (1) we study *self*-verification, where the same parallel reasoner model judges its own outputs. This eliminates the need for additional training with curated judge data, saves memory, and reduces the computational overhead associated with external verification. (2) We co-train the model to be both a solution generator and a pairwise self-verifier in a unified framework.

**Co-training for generation and verification.** Training separate verifiers incurs significant overhead in compute, memory, and data curation. Recent work has explored co-training a single model for both generation and verification. Sareen et al. (2025) train models to self-verify online during RL, ensuring the verifier sees in-distribution generations and uses this capability to perform better majority voting at test-time for math reasoning. Liu et al. (2025a) extends this to unified CoT verifiers, while Zha et al. (2025) co-trains a separate process reward model that provides intermediate rewards. Offline approaches train verifiers or aggregators on data sampled from the base model (Venkatraman et al., 2025; Ruan et al., 2025). For code generation specifically, several works co-train generators with unit-test producers (Wang et al., 2025; Lee et al., 2025). However, existing co-training methods rely on *pointwise* verification rewards. Our $V_1$-PairRL framework instead co-trains for *pairwise* self-verification in an online, co-evolving setup, which we show yields superior test-time scaling performance.

## B. V₁-PointRL Training Details

To provide a fair comparison for our pairwise verification approach, we train $\mathbf{V}_1$-PointRL using the same co-evolving setup as $\mathbf{V}_1$-PairRL. The key difference lies in the verification objective: instead of comparing pairs of solutions, $\mathbf{V}_1$-PointRL assigns an independent score to each solution.

**Verification Reward.** For pointwise verification, the model evaluates a single solution $s$ and outputs a confidence score between 1-10 which is normalized to $v \in [0, 1]$. The reward is computed as:

$$r_{\text{verif}} = \mathbb{I}(|v - y| \leq 0.2) \cdot (1 - |v - y|) \tag{5}$$

where $y \in \{0, 1\}$ is the ground truth correctness. The sparsity threshold prevents the "safe bet" collapse mode where the model learns to always output $v = 0.5$ (in practice, not applying the sparsity threshold leads to collapse in solution generation performance and is necessary for stable training).

**Training Configuration.** All hyperparameters (learning rate, batch size, rollout allocation, etc.) are identical to $\mathbf{V}_1$-PairRL as shown in Table 4. The only difference is the verification prompt and reward computation.

**Inference.** At inference time, $\mathbf{V}_1$-PointRL generates $N$ solutions and independently scores each one. The solution with the highest pointwise score is selected as the final answer.

## C. Training Setup Details for Code, Math, and Rubric-based RL

This section consolidates the per-domain training configurations used in Section 4. The main paper summarizes these in a single paragraph; full details are provided here.

**Training Setup for Code RL.** We adopt the DAPO (Yu et al., 2025) configuration, removing the KL penalty, using Clip High, and applying token-level loss, and follow Dr. GRPO (Liu et al., 2025b) by removing standard deviation normalization. To ensure a fair comparison, we enforce a fixed compute budget of 8 total rollouts per problem. The baseline allocates all 8 rollouts to the solver, while co-evolving models ($\mathbf{V}_1$-PairRL and $\mathbf{V}_1$-PointRL) split this budget into 4 solver and 4 verifier rollouts. We verified that training the baseline for a longer duration with fewer rollouts does not improve performance, confirming that gains stem from the co-training objective. All models are trained for 150 steps, with checkpoints selected based on validation accuracy (Pass@1). Prompts used for code-generation training are in Section J.

**Training Setup for Math RL.** We adopt the recipe of Sareen et al. (2025) and train `Qwen3-4B-Base` on the Polaris-53K math dataset (An et al., 2025) for ∼300 steps with GRPO. Whereas Sareen et al. (2025) train the pointwise verifier with an SFT loss, our $\mathbf{V}_1$-PointRL baseline instead trains a *generative* pointwise verifier end-to-end with RL; $\mathbf{V}_1$-PairRL then replaces this pointwise RL loss with the pairwise self-verification loss of Section 4. We evaluate on MATH-Hard and AIME 2025 at $N$=16 candidate solutions.

**Training Setup for Rubric-based RL (non-verifiable).** To test $\mathbf{V}_1$-PairRL outside of verifiable domains, we train `Qwen3-4B-Instruct` on the RaR-Medicine dataset (Gunjal et al., 2025) (17,926 medical QA prompts with physician-curated rubrics) for 100 steps with GRPO, using rubric-graded scores from an LLM judge as the reward in place of test-case correctness; the same scores label correct/incorrect solutions for the pairwise verifier during co-training. The trained policy pairwise self-verifies its own solutions *without* access to dataset rubrics. We evaluate Pass@1 with self-verification on HealthBench Main (Arora et al., 2025) and a held-out RaR-Medicine test split, with GPT-5-nano as the grader.

*Configuration.* Base model: `Qwen3-4B-Instruct-2507`. Training data: RaR-Medicine (Gunjal et al., 2025) (17,926 medical QA problems). Algorithm: GRPO with 8 solver rollouts per problem and 8 judge rollouts for co-training, run for 100 training steps in the Trinity-RFT framework (veRL + Ray backend).

*Generator (solver) reward.* The external judge is `Qwen3-30B-A3B-Instruct` (frozen, temperature = 0). For each candidate response $o$, the judge rates the response against the dataset rubric on a 1–10 scale and we normalise the rating to $s(o) \in [0, 1]$. This continuous score $s(o)$ is the generator reward fed to GRPO; no threshold or sparsity gate is applied on the solver side.

*Self-verification (verifier) reward.* The verifier is trained to imitate the external judge's scores. For $\mathbf{V}_1$-PointRL the policy rates a single solution on a 1–10 scale (call the normalised value $v_i \in [0, 1]$); the verifier reward is $1 - |v_i - y_i|$, where $y_i = s(o_i)$ is the external-judge score. For $\mathbf{V}_1$-PairRL the policy rates both members of a pair $(A, B)$ and the verifier reward is $1 - \frac{1}{2}(\text{MAE}_A + \text{MAE}_B)$ with $\text{MAE}_i = |v_i - y_i|$. In both cases we apply the same *sparsity threshold* at 0.7, so that the

actual reward delivered to the optimiser is

$$r_{\text{verif}} = \begin{cases} 1 - \text{MAE} & \text{if } 1 - \text{MAE} \geq 0.7, \\ 0 & \text{otherwise,} \end{cases} \qquad (6)$$

which is the rubric-RL analogue of the indicator-based sparsity used in code-RL (Eq. 5 in App. B) and prevents the verifier from receiving credit for low-quality MAE-$> 0.3$ predictions.

*Pair formation ($V_1$-PairRL only).* The pair builder reuses the same $0.7$ value as a *quality split*: rollouts with $s \geq 0.7$ are labelled "high quality" and rollouts with $s < 0.7$ "low quality". Mixed (high $\times$ low) pairs are formed first and prioritised; if more pairs are needed up to the per-step cap, the builder falls back to high-only and then low-only pairs (with the low-only count capped at the high-only count) and randomises which member of each pair is assigned position $A$ vs. $B$. This is the rubric-RL realisation of the Correct–Incorrect / Correct–Correct pairing strategy of §4.

*Evaluation scoring.* On **HealthBench Main**, the rubric score for a response is $\sum(\text{achieved rubric points})/\sum(\text{positive possible points})$, where each physician-written criterion is graded as binary satisfied / not satisfied by GPT-5-nano; the score range therefore includes negative values when safety-penalty criteria are triggered. On the held-out **RaR-Medicine** test split we use the identical aggregation formula but with synthetically generated rubric items (per-item weights ranging from $-2$ to $+5$), again graded by GPT-5-nano; scores typically fall in $[0, 1]$.

## D. Co-evolving Training Ablation

$\mathbf{V}_1$-PairRL trains solution generation and verification capabilities in a *co-evolving* setup, where verification samples are drawn online from the same iteration of the generator that produced them. This ensures the verifier always trains on in-distribution outputs of the current policy. An alternative is to jointly optimize generation and verification in a multi-task setup where the verification data is generated offline, using the *base* model, before training.

To isolate the effect of co-evolving training, we train a *non-co-evolving* baseline: the same model trained with RL for both generation and verification, but where the verifier prompt contains 8 offline solutions per problem generated once by Qwen3-4B-Instruct-2507 on the DeepCoder prompts, rather than online solutions from the evolving policy. We apply $\mathbf{V}_1$-Infer at inference time with the same $2\times$ budget for both setups. As shown in Figure 10, co-evolving training consistently outperforms the non-co-evolving setup across all code benchmarks, confirming that adapting the verifier to the generator's evolving distribution is a key driver of $\mathbf{V}_1$-PairRL's gains.

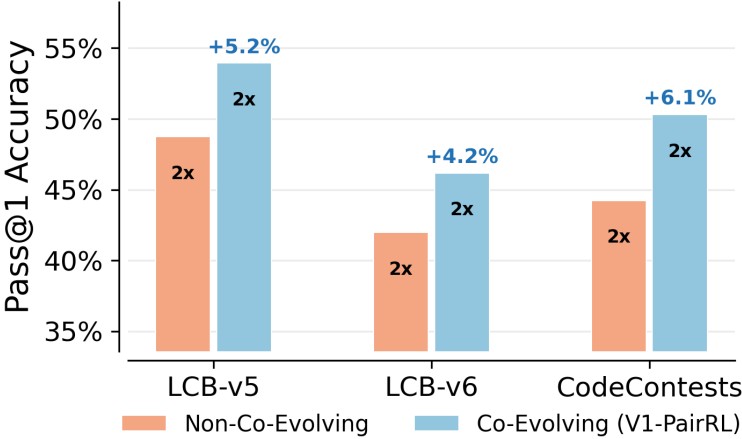

*Figure 10.* Co-evolving vs. non-co-evolving pairwise verification. Both use $\mathbf{V}_1$-Infer ($2\times$ budget).

# E. Hyperparameters

## E.1. Inference Sampling Parameters

**Inference Framework.** We use SGLang (Zheng et al., 2024) for all inference experiments, which provides efficient batched inference for large language models.

**Sampling Parameters.** We set temperature $T = 0.6$ for all code generation experiments and $T = 1.0$ for all math reasoning experiments to encourage better exploration of the solution space. We use top-p sampling with $p = 0.95$ for all experiments. For experiments with **V$_1$**-Infer in Section 3 where we test the inference algorithm on widely used open source LLMs like GPT-OSS-20B, GPT-OSS-120B, Qwen3-4B-Instruct-2507, Qwen3-4B-Thinking-2507, we set the max generation length to 32768 for all methods.

For trained model evaluations in Section 4 we find that training all models leads to longer chains of thoughts, as is commonly the case in RLVR (DeepSeek-AI, 2025). This leads to frequent truncation, especially for the baseline RL model. To resolve this, we adopt the truncate-and-continue-generation method from prior work (Yang et al., 2025; Sareen et al., 2025). When generation reaches the maximum token budget (`finish_reason=length`), we resume decoding by appending the partially generated assistant output to the message history and issuing a continuation request. If the truncated output does not contain a closing `</thinking>` tag, we append the following transition sentence before continuing: *"Considering the limited time by the user, I have to directly give the required response based on the reasoning till now directly.`</thinking>`"*. This explicitly closes the thinking block and forces the continuation to produce the final answer. If the truncated output already contains `</thinking>`, we directly continue generation without appending any additional text. We continue the generation for 2K tokens, making the total maximum generation length 34 K for all experiments in Section 4.

**Number of Samples.** We generate $N \in \{8, 16\}$ candidate solutions for most experiments. For budget-matched comparisons between pointwise and pairwise verification, we additionally evaluate pointwise verification with $N = 32$ samples to match the total LLM calls of pairwise verification with $N = 16$ and budget $3\times$. This budget-matched comparison is performed only for GPT-OSS-20B and Qwen3-4B-Instruct models.

**Verification Budget.** For **V$_1$**-Infer experiments in Section 4, we evaluate verification budgets of $1\times$, $2\times$, and $3\times$ the number of base solutions ($N$). Full results across all budgets are provided in the appendix figures.

**Experimental Runs.** All inference experiments for base models (Section 4) are run once. All trained model results (Section 5) are run 3 times with different random seeds, and we report the mean. Due to the computational cost of running multiple seeds, we limit the budget exploration for trained models to $1\times$ and $2\times$, though we believe performance can further scale with additional budget for the pairwise method (**V$_1$**-Infer).

**Swiss Refinement Algorithm Parameters.** For **V$_1$**-Infer, we use the following settings: Minimum degree $d_{\min} = 2$ (each solution compared at least twice in coverage phase), Swiss window size $h = 8$, Confidence floor set to a small value, $\tau = 0.1$ for weighted aggregation. We do not tune parameters to any specific benchmark. We use the same parameters across benchmarks or models across all experiments.

## E.2. Training Hyperparameters

Table 4 summarizes the key training hyperparameters. We train models using rLLM (Tan et al., 2025) and verl (Sheng et al., 2024) backend.

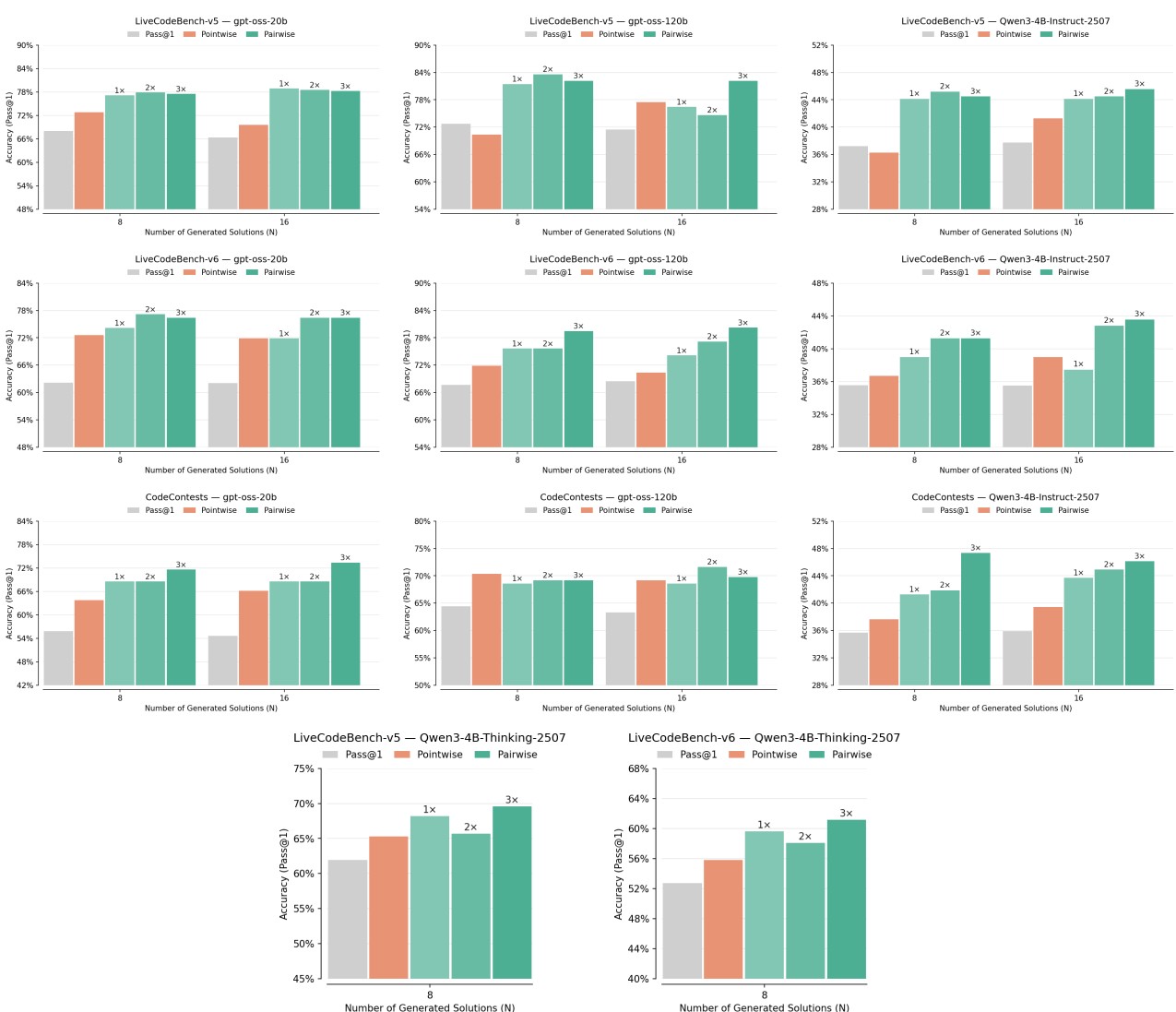

*Figure 11.* Self-verification bar plots across benchmarks and models (LiveCodeBench-v5, LiveCodeBench-v6, and CodeContests). Each plot compares Pass@1, pointwise verification, and pairwise verification (budgets $1\times/2\times/3\times$) at $N \in \{8, 16\}$. For Qwen3-4B-Thinking (bottom row), we run experiments only on LiveCodeBench at $N = 8$ to manage compute.

*Table 4.* Training hyperparameters for $\mathbf{V}_1$ on Code-RL.

| Hyperparameter | Qwen3-4B-Inst |
|---|---|
| Learning rate | $1 \times 10^{-6}$ |
| Batch size | 64 |
| Rollouts per prompt ($G$) | 4 (8 for baseline RL) |
| Judge rollouts | 4 (0 for baseline RL) |
| Max prompt length | 10240 |
| Max response length | 24576 |
| Temperature | 0.6 |
| Top-p | 0.95 |
| Clip ratio (low) | 0.2 |
| Clip ratio (high) | 0.28 |
| $\lambda$ (loss weight) | 1.0 |
| KL coefficient | 0 |
| Entropy coefficient | 0 |
| Std normalization | No |
| Loss aggregation | token-mean |

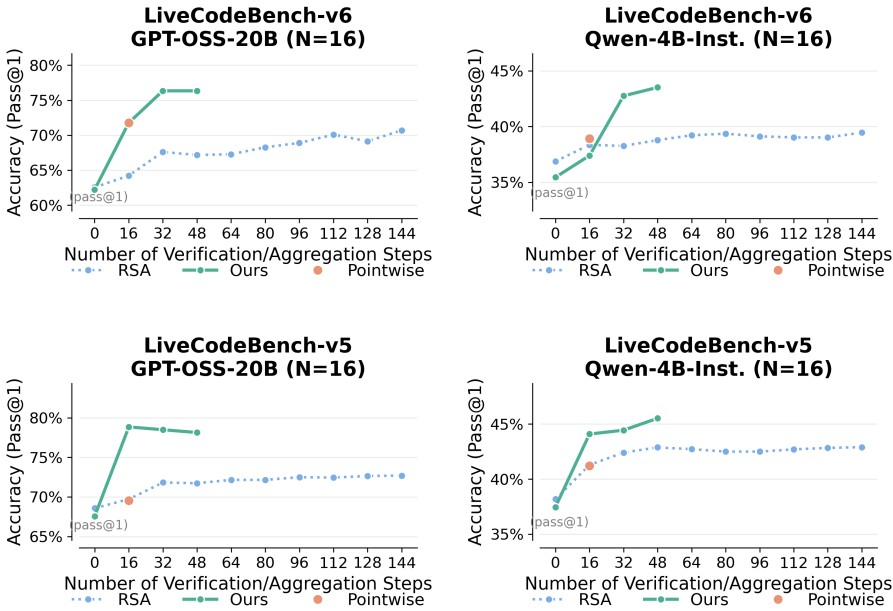

*Figure 12.* **Comparing Pairwise Verification with RSA.** Top row: LiveCodeBench-v6. Bottom row: LiveCodeBench-v5. Columns (left→right): gpt-oss-20b, Qwen3-4B-Instruct-2507. **Here number of generations by the base model is 16**, followed by verification (pointwise, pairwise) or aggregation (RSA).

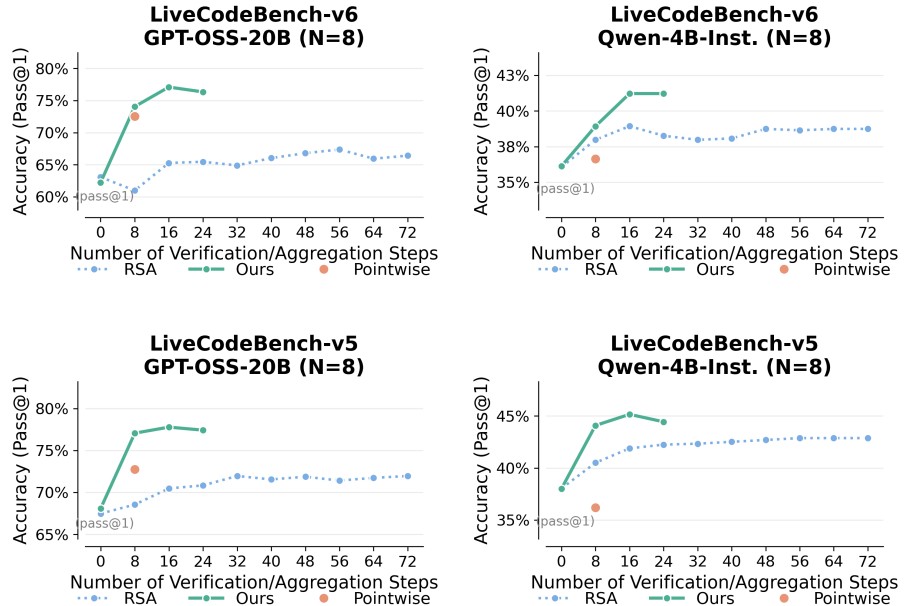

*Figure 13.* **Comparing Pairwise Verification with RSA.** Top row: LiveCodeBench-v6. Bottom row: LiveCodeBench-v5. Columns (left→right): gpt-oss-20b, Qwen3-4B-Instruct-2507. **Here number of generations by the base model is 8**, followed by verification (pointwise, pairwise) or aggregation (RSA).

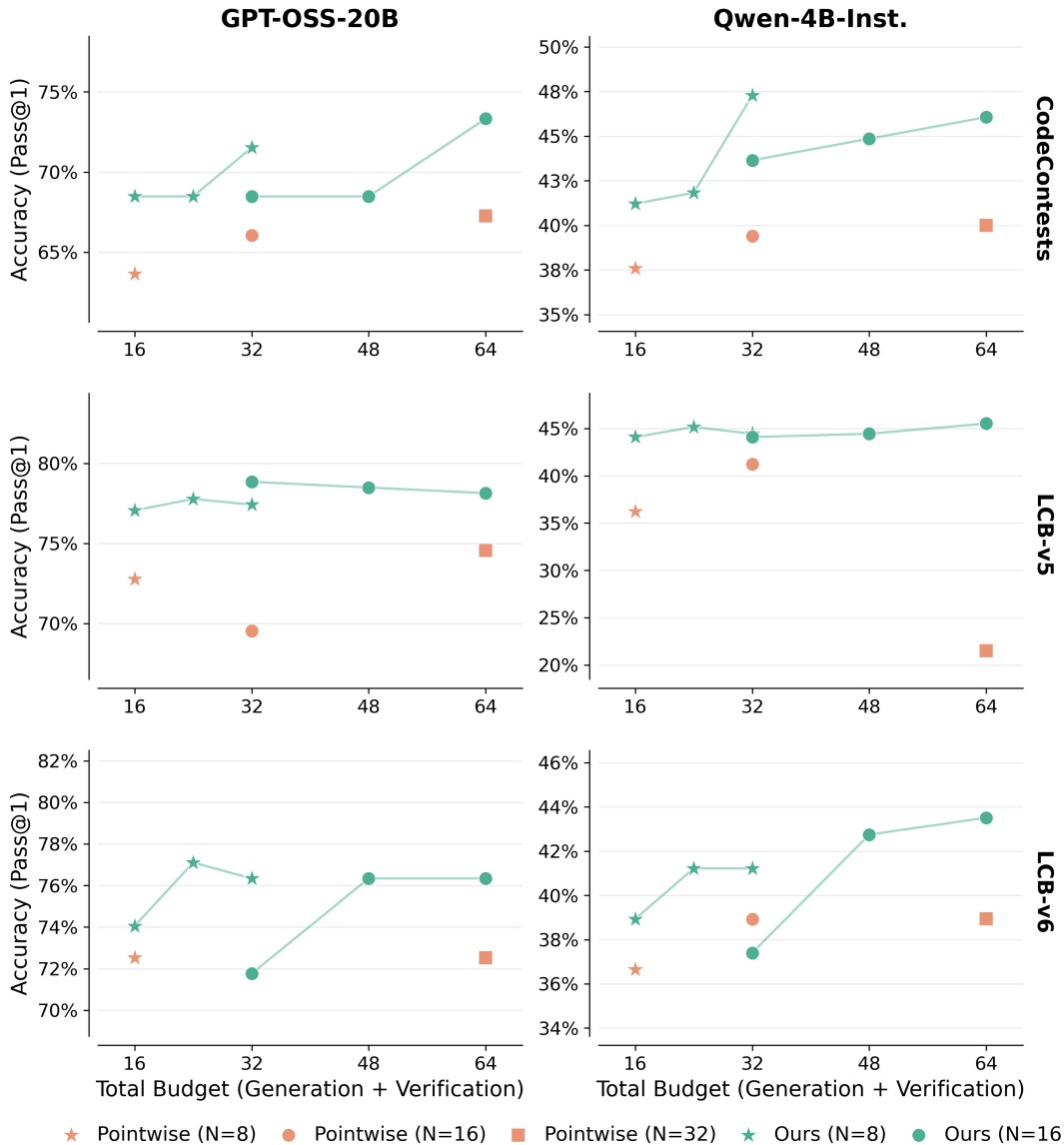

*Figure 14.* **Budget vs accuracy per benchmark.** Accuracy vs. total budget (generation + verification calls) for CodeContests (top row), LiveCodeBench-v5 (middle row), and LiveCodeBench-v6 (bottom row). Left column: GPT-OSS-20B. Right column: Qwen3-4B-Inst. Stars denote $N = 8$ base generations, circles denote $N = 16$, squares denote $N = 32$ (pointwise only).

# F. Detailed Algorithm

This section provides the full specification of the helper procedures used in Algorithm 1. The main paper presents a high-level view for clarity; here we give the exact update rules and pairing strategies required for reproducibility.

**State.** We maintain a state $\mathcal{T} = (\{\mu_i\}_{i=1}^N, \{d_i\}_{i=1}^N, \mathcal{H})$, where $\mu_i$ is the current estimated quality score of solution $s_i$, $d_i$ is the number of distinct opponents it has been compared against, and $\mathcal{H} \subset [N] \times [N]$ records previously compared pairs.

**CoveragePairs.** Ensures minimum degree while anchoring comparisons to similar-quality peers.

---
**Algorithm 2** COVERAGEPAIRS($\mathcal{T}, d_{\min}$)
---
**Require:** State $\mathcal{T} = (\{\mu_i\}, \{d_i\}, \mathcal{H})$, minimum degree $d_{\min}$
**Ensure:** Disjoint pair set $\mathcal{P}$
1: $\mathcal{P} \leftarrow \emptyset$, $\mathcal{U} \leftarrow \emptyset$ {used indices}
2: $\mathcal{L} \leftarrow \{i : d_i < d_{\min}\}$; sort $\mathcal{L}$ by increasing $d_i$
3: **for** each $i \in \mathcal{L}$ **do**
4:     **if** $i \in \mathcal{U}$ **then**
5:         **continue**
6:     **end if**
7:     $\mathcal{C} \leftarrow \{j \neq i : j \notin \mathcal{U}, (i,j) \notin \mathcal{H}\}$
8:     **if** $\mathcal{C} = \emptyset$ **then**
9:         **continue**
10:     **end if**
11:     $j \leftarrow \arg\min_{j \in \mathcal{C}} |\mu_i - \mu_j|$
12:     $\mathcal{P} \leftarrow \mathcal{P} \cup \{(i,j)\}$
13:     $\mathcal{U} \leftarrow \mathcal{U} \cup \{i,j\}$
14: **end for**
15:
16: **return** $\mathcal{P}$

---

**UpdateStats.** This is where the mathematical aggregation is implemented. Note. In practice, the update of $\mu_i$ is

---
**Algorithm 3** UPDATESTATS($\mathcal{T}, \mathcal{O}, \tau$)
---
**Require:** State $\mathcal{T} = (\{\mu_i\}, \{d_i\}, \mathcal{H})$, outcomes $\mathcal{O}$, floor $\tau > 0$
**Ensure:** Updated state $\mathcal{T}$
1: **for** each $(i, j, w, r_i, r_j) \in \mathcal{O}$ **do**
2:     $v_{ij} \leftarrow \mathbb{I}[w = \texttt{A}] + 0.5 \cdot \mathbb{I}[w = \texttt{tie}]$
3:     $w_{ij} \leftarrow \max\left(\frac{|r_i - r_j|}{9}, \tau\right)$
4:     update $\mu_i, \mu_j$ using Eq. (1)
5:     **if** $(i,j) \notin \mathcal{H}$ **then**
6:         $d_i \leftarrow d_i + 1$;   $d_j \leftarrow d_j + 1$
7:         $\mathcal{H} \leftarrow \mathcal{H} \cup \{(i,j), (j,i)\}$
8:     **end if**
9: **end for**

---

implemented incrementally using sufficient statistics, but Eq. (1) fully specifies the aggregation semantics.

**SwissPairs.** Allocates remaining budget to the most informative near-ties.

The above procedures jointly ensure (i) early stabilization of the comparison topology via minimum-degree coverage and (ii) efficient use of remaining budget through uncertainty-focused Swiss refinement, enabling accurate ranking with $O(N)$–scale pairwise verification.

**Algorithm 4** SWISSPAIRS($\pi, \mathcal{T}, h$)

**Require:** Ranking $\pi$, state $\mathcal{T} = (\{\mu_i\}, \{d_i\}, \mathcal{H})$, window size $h$
**Ensure:** Disjoint pair set $\mathcal{P}$
1: $\mathcal{P} \leftarrow \emptyset$, $\mathcal{U} \leftarrow \emptyset$
2: **for** $k = 1$ **to** $N$ **do**
3:     $i \leftarrow \pi_k$
4:     **if** $i \in \mathcal{U}$ **then**
5:         **continue**
6:     **end if**
7:     $\mathcal{C} \leftarrow \emptyset$
8:     **for** $t = k + 1$ **to** $\min(k + h, N)$ **do**
9:         $j \leftarrow \pi_t$
10:         **if** $j \in \mathcal{U}$ **then**
11:             **continue**
12:         **end if**
13:         add $(|\mu_i - \mu_j|, \mathbb{I}[(i,j) \in \mathcal{H}], j)$ to $\mathcal{C}$
14:     **end for**
15:     **if** $\mathcal{C} \neq \emptyset$ **then**
16:         choose $j$ with lexicographically minimal key in $\mathcal{C}$
17:         $\mathcal{P} \leftarrow \mathcal{P} \cup \{(i,j)\}$
18:         $\mathcal{U} \leftarrow \mathcal{U} \cup \{i, j\}$
19:     **end if**
20: **end for**
21:
22: **return** $\mathcal{P}$

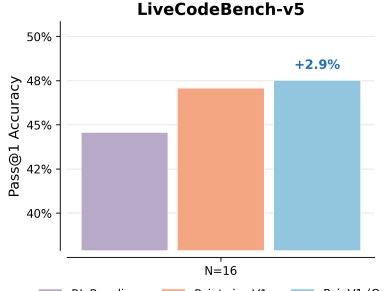 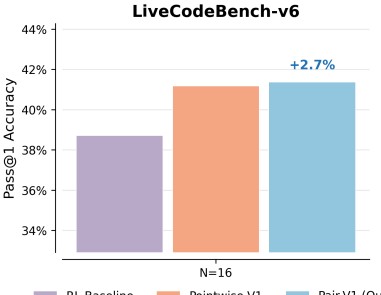 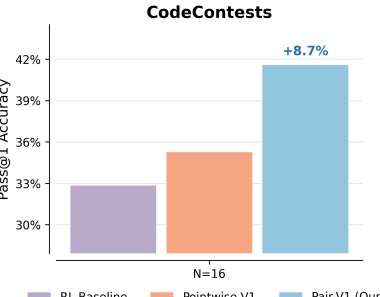

*Figure 15.* $V_1$**-PairRL vs RL baseline across benchmarks (N=16).** Left: LiveCodeBench-v5. Middle: LiveCodeBench-v6. Right: CodeContests. Co-training with pairwise verification consistently improves Pass@1 accuracy across all benchmarks.

# G. Additional $\mathbf{V}_1$-Infer Analyses

This appendix collects three supporting analyses for $\mathbf{V}_1$-Infer (Section 3) that did not fit in the main paper: the random-pair baseline, comparison against external reward / process reward models, and a length-bias analysis on HealthBench.

## G.1. Uncertainty-Guided Selection vs. Random-Pair Baseline

To validate the effectiveness of our uncertainty-guided refinement algorithm, we compare against a baseline that randomly selects pairs for pairwise comparison. On LCB-v6 with GPT-OSS-20B at budget $3\times$, $\mathbf{V}_1$-Infer achieves 76.3% accuracy compared to 72.5% for random pairing, a gain of $+3.8\%$. This demonstrates that the strategic pair selection in $\mathbf{V}_1$-Infer, which prioritizes comparisons between solutions with similar quality scores, yields more informative judgments than random sampling.

## G.2. Comparison with External Reward Models and Process Reward Models

A natural alternative to self-verification is to select among the $N$ candidate solutions with a separately trained reward model. We compare $\mathbf{V}_1$-Infer against four publicly released math-tuned reward / process reward models using best-of-$N$ selection on AIME 2025 ($N{=}16$): AceMath-7B-RM (Liu et al., 2024), Skywork-o1-Open-PRM-Qwen-2.5-7B (He et al., 2024), Qwen2.5-Math-PRM-7B (Zhang et al., 2025b), and ThinkPRM-7B (Khalifa et al., 2025) (a generative, "thinking" PRM). Table 5 reports accuracy (%) across three off-the-shelf base generators.

*Table 5.* Best-of-$N$ selection on AIME 2025 ($N{=}16$). External (Ext.) ORM/PRM scoring is compared against self-verification with $\mathbf{V}_1$-Infer. Pointwise and $\mathbf{V}_1$-Infer numbers are reported as mean±std over 3 seeds.

| Method | Type | gpt-oss-120b | gpt-oss-20b | qwen3-4b |
|---|---|---|---|---|
| Pass@1 (random) | – | 76.0 | 65.2 | 49.2 |
| BoN AceMath-7B-RM | Ext. ORM | 60.0 | 63.3 | 43.3 |
| BoN Skywork-o1-PRM | Ext. PRM | 73.3 | 60.0 | 36.7 |
| BoN Qwen2.5-Math-PRM-7B | Ext. PRM | 76.7 | 56.7 | 56.7 |
| BoN ThinkPRM-7B | Ext. Gen. PRM | 73.3 | 70.0 | 40.0 |
| Pointwise self-verification | Self-verify | 85.6±5.10 | 85.6±1.96 | 54.5±6.93 |
| **$\mathbf{V}_1$-Infer (Ours)** | Self-verify | **92.2±1.91** | **87.8±1.91** | **62.2±1.91** |

Across all three base generators, externally trained ORMs/PRMs underperform self-verification with the same base model. We attribute this to a distribution mismatch between the reward models' training data (typically older Qwen-family outputs) and the generators evaluated here. $\mathbf{V}_1$-Infer consistently outperforms all four external verifiers and the pointwise self-verification baseline, with substantially lower variance than pointwise across seeds.

## G.3. Pairwise Verification is Less Length-Biased than Pointwise (HealthBench)

A common concern in non-verifiable settings is that LLM judges may favor superficially fluent or longer answers. We compute the Spearman rank correlation between each verifier's score and the response length on HealthBench; lower magnitude indicates less length bias. Across all six model-benchmark cells in Table 6, $\mathbf{V}_1$-Infer's correlation is lower than pointwise verification's, showing that pairwise comparison provides a better calibrated signal than independent pointwise scoring even in non-verifiable domains.

*Table 6.* Spearman correlation between verifier score and response length on HealthBench (lower magnitude is better, i.e., less length-biased).

| Benchmark | Model | $\mathbf{V}_1$-Infer vs. Length | Pointwise vs. Length |
|---|---|---|---|
| HealthBench Hard | gpt-oss-20b | **0.330** | 0.373 |
| | gpt-oss-120b | **0.189** | 0.236 |
| | Qwen3-4B-Instruct | **0.086** | 0.088 |
| HealthBench Main | gpt-oss-20b | **0.218** | 0.280 |
| | gpt-oss-120b | **0.146** | 0.182 |
| | Qwen3-4B-Instruct | **0.086** | 0.118 |

## H. Multi-Seed Inference Results

To verify that the test-time scaling gains reported in Section 3 are robust to sampling variance, we re-run the main inference experiments with 3 random seeds for each model$\times$benchmark combination ($N$=16 base generations per seed). Table 7 reports mean $\pm$ standard deviation. $\mathbf{V}_1$-Infer outperforms both Pass@1 and pointwise self-verification across all seeds and settings, and matches or exceeds RSA in 6 out of 7 settings. Critically, $\mathbf{V}_1$-Infer also exhibits lower variance than pointwise in 6 of the 7 settings (e.g., std $0.87$ vs. $3.77$ on `gpt-oss-20b` LCBv6; std $1.91$ vs. $5.10$ on `gpt-oss-120b` AIME), indicating that pairwise self-verification provides a more stable signal than independent scoring.

*Table 7.* Multi-seed results (3 seeds, $N$=16). Mean $\pm$ std accuracy (%). Best per row in bold.

| Model | Benchmark | Pass@1 | Pointwise | RSA $K$=4, $T$=9 | $\mathbf{V}_1$-Infer (Ours) |
|---|---|---|---|---|---|
| gpt-oss-120b | AIME 2025 | $76.4 \pm 1.00$ | $85.6 \pm 5.10$ | $88.9 \pm 1.91$ | $\mathbf{92.2 \pm 1.91}$ |
| | LCBv6 | $68.8 \pm 0.46$ | $74.6 \pm 1.17$ | $76.1 \pm 0.87$ | $\mathbf{79.4 \pm 0.00}$ |
| gpt-oss-20b | AIME 2025 | $65.8 \pm 0.65$ | $85.6 \pm 1.96$ | $85.6 \pm 1.96$ | $\mathbf{87.8 \pm 1.91}$ |
| | LCBv6 | $60.3 \pm 0.85$ | $72.2 \pm 3.77$ | $69.0 \pm 2.71$ | $\mathbf{75.8 \pm 0.87}$ |
| Qwen3-4B-Instruct | AIME 2025 | $47.7 \pm 1.05$ | $54.5 \pm 6.93$ | $\mathbf{70.0 \pm 0.00}$ | $62.2 \pm 1.91$ |
| | LCBv6 | $34.7 \pm 1.36$ | $37.2 \pm 2.91$ | $38.7 \pm 0.87$ | $\mathbf{43.2 \pm 0.47}$ |
| Qwen3-4B-Thinking | LCBv6 | $52.8 \pm 0.42$ | $58.4 \pm 0.57$ | $52.7 \pm 0.00$ | $\mathbf{63.4 \pm 2.19}$ |

## I. Token-Level Efficiency of $\mathbf{V}_1$-Infer

Pairwise prompts can consume more tokens per call than pointwise prompts. We therefore report *effective tokens* $=$ output tokens $+ 0.2 \cdot$ input tokens, motivated by the $4$–$8\times$ price differential between prefill and decode tokens in standard API pricing (e.g., GPT-4o-class APIs). Table 8 reports total effective tokens (averaged over 3 seeds) for `gpt-oss-120b` at $N$=16 on AIME 2025 and LiveCodeBench-v6.

*Table 8.* Token-level efficiency on AIME 2025 and LiveCodeBench-v6 with `gpt-oss-120b` ($N$=16, 3 seeds). Effective tokens are dataset totals.

| Method | AIME Acc. | AIME Eff. Tokens | LCBv6 Acc. | LCBv6 Eff. Tokens |
|---|---|---|---|---|
| Pointwise $N$=32 (64 calls) | 83.3 | 7.7M | 75.6 | 27.5M |
| RSA $K$=4, $T$=9 | 88.9 | 17.0M | 76.1 | 87.0M |
| $\mathbf{V}_1$-**Infer** $N$=16 **3**$\times$ **(64 calls)** | **92.2** | **6.8M** | **79.4** | 31.6M |

$\mathbf{V}_1$-Infer achieves higher accuracy than pointwise $N$=32 at lower token cost on AIME ($0.88\times$) and comparable cost on LCBv6 ($1.15\times$), while being $2.0$–$2.7\times$ cheaper than RSA at higher accuracy. The same trend holds for `gpt-oss-20b` and `Qwen3-4B-Instruct`: $\mathbf{V}_1$-Infer dominates pointwise $N$=32 on both accuracy and effective tokens, and beats RSA at $2.0$–$2.4\times$ lower cost.

# J. Generation Prompts

## J.1. Code Generation

---
**Code Generation Prompt**

```
**Problem**
{problem}

Think and reason step by step before coding the final solution for the problem above.
Put your reasoning and any draft coding solutions between <thinking> ...  </thinking>
tags.  After the reasoning (i.e.  after the </thinking> tag), use the format provided
in the problem above (code-block with backticks) to format your final code solution.
Do not include any thinking within the code block.

### Answer:
```
---

Note that "Do not include any thinking within the code-block." is added to the prompt, since in the absence of it, training the model (Qwen3-4B-Instruct-2507) leads to it generating large amounts of comments within the code block, effectively leading to truncation of the response (due to reaching max length) and hence, 0 rewards and training collapse.

## J.2. Math Solution Generation

---
**Math Solution Generation Prompt**

```
{problem}.  Let's think step by step and output the final answer within \boxed{}.
```
---

# K. Verification Prompts

This section provides the exact prompts used for pointwise and pairwise self-verification in our experiments. Both verification methods use a 1–10 rating scale for fair comparison.

## K.1. Pointwise Verification Prompts

### K.1.1. CODE GENERATION

> **Pointwise Code Verification Prompt**
>
> ```
> You are an expert code reviewer.  Rate the correctness of a solution to a programming
> problem.
>
> **Evaluation Guidelines:**
> - Analyze the problem's requirements and constraints.
> - Mentally trace the solution with test cases (including edge cases) to verify
> correctness.
> - Give a higher score if the solution is robust and fault-tolerant.
>
> **Problem**
> {problem}
>
> **Solution**
> {code}
>
> **Output Format:**
> First, provide your step-by-step reasoning.  Then, on a new line, provide your final
> rating using the EXACT tags below.  Add no other text after the tags.
> <rating>INTEGER_1_TO_10</rating>
>
> **Rating Rules:**
> - Rate correctness on a 1-10 scale (10 = correct & robust, 5 = borderline, 1 =
> incorrect).
> Please provide your analysis now.
> ```

### K.1.2. MATH REASONING

> **Pointwise Math Verification Prompt**
>
> ```
> You are an expert math contest grader.  Rate the correctness of a submission based
> solely on the final answer.
>
> **Evaluation Guidelines:**
> - Extract the submission's final answer.  Use any provided reasoning only to help you
> assess whether the stated final answer is trustworthy.  Do not award credit for method
> quality or rigor.
> - Carefully analyze the problem statement and the submission to assess whether the
> final answer is correct.  Grade only the final answer.
>
> **Problem**
> {problem}
>
> **Solution**
> {solution}
>
> **Output Format:**
> First, provide your reasoning (what checks you performed).  Then, on a new line,
> provide your final rating using the EXACT tag below.  Add no other text after the tag.
> <rating>INTEGER_1_TO_10</rating>
>
> **Rating Rules:**
> - Rate correctness on a 1-10 scale (10 = certainly correct, 8 = very likely correct, 5
> = uncertain/borderline, 3 = likely incorrect, 1 = certainly incorrect).
> Please provide your analysis now.
> ```

## K.2. Pairwise Verification Prompts

### K.2.1. CODE GENERATION

---

**Pairwise Code Verification Prompt**

You are an expert code reviewer.  Compare two solutions to a programming problem and rate their correctness.

**Evaluation Guidelines:**
- Analyze the problem's requirements and constraints.
- Mentally trace each solution with test cases (including edge cases) to verify correctness.
- If both solutions appear correct, prefer the more robust and fault-tolerant one.

**Problem**
{problem}

**Solution A**
{code_A}

**Solution B**
{code_B}

**Output Format:**
First, provide your step-by-step reasoning.  Then, on separate new lines, provide your final ratings using the EXACT tags below.  Add no other text after the tags.
<rating_A>INTEGER_1_TO_10</rating_A>
<rating_B>INTEGER_1_TO_10</rating_B>

**Rating Rules:**
- Rate correctness on a 1-10 scale (10 = correct & robust, 5 = borderline, 1 = incorrect).
- The higher rating wins.  Equal ratings imply a tie.
Please provide your analysis now.

---

### K.2.2. MATH REASONING

---

**Pairwise Math Verification Prompt**

You are an expert math contest grader.  Compare two submissions and rate correctness based solely on the final answer.

**Evaluation Guidelines:**
- Extract each submission's final answer.  Use any provided reasoning only to help you assess whether the stated final answer is trustworthy.  Do not award credit for method quality or rigor.
- Carefully analyze the problem statement and the submissions to assess whether each final answer is correct.  Grade only the final answer.

**Problem**
{problem}

**Solution A**
{sol_A}

**Solution B**
{sol_B}

**Output Format:**
First, provide your reasoning (what checks you performed).  Then, on separate new lines, give ratings using the EXACT tags below.  Add no other text after the tags.
<rating_A>INTEGER_1_TO_10</rating_A>
<rating_B>INTEGER_1_TO_10</rating_B>

**Rating Rules:**
- Rate correctness on a 1-10 scale (10 = certainly correct, 8 = very likely correct, 5 = uncertain/borderline, 3 = likely incorrect, 1 = certainly incorrect).
- Higher rating wins.  Equal ratings imply a tie.
Please provide your analysis now.

---

# L. SWE-bench Verification Examples

This section presents representative examples from our SWE-bench Lite evaluation (300 instances, 8 candidate patches each, Gemini 2.5 Flash as both generator and verifier). We show cases where pairwise and pointwise verification select different patches, illustrating how head-to-head comparison can surface correct solutions that independent scoring misses, and vice versa. For each example, we show the issue description, the patches selected by each method, and their rankings.

## L.1. Example 1: `django__django-11049`: Pairwise Correct, Pointwise Wrong

> **Issue: Correct expected format in invalid DurationField error message**
>
> If you enter a duration "14:00" into a duration field, it translates to "00:14:00" which is 14 minutes. The current error message for invalid DurationField says that this should be the format of durations: `[DD] [HH:[MM:]]ss[.uuuuuu]`. But according to the actual behaviour, it should be: `[DD] [[HH:]MM:]ss[.uuuuuu]`, because seconds are mandatory, minutes are optional, and hours are optional if minutes are provided. This seems to be a mistake in all Django versions that support the DurationField. Also the duration fields could have a default `help_text` with the requested format, because the syntax is not self-explanatory.

**Setup:** 8 candidate patches generated, all non-empty.
**Pairwise ranking:** [6, 3, 5, 1, 7, 0, 4, 2]   → selected **idx=6** ✓
**Pointwise ranking:** [0, 1, 3, 5, 6, 7, 4, 2]   → selected **idx=0** ✗

> **Pairwise-Selected Patch (idx=6): Resolves the issue ✓**
>
> ```
> diff --git a/django/db/models/fields/__init__.py
> @@ -1587,7 +1587,7 @@ class DurationField(Field):
> default_error_messages = {
> 'invalid': _("'%(value)s' value has an invalid format.  It must be in "
> -         "[DD] [HH:[MM:]]ss[.uuuuuu] format.")
> +         "[DD] [[HH:]MM:]ss[.uuuuuu] format.")
> }
> diff --git a/django/forms/fields.py
> @@ -467,6 +467,8 @@ class DurationField(Field):
> +  help_text = _('Accepted format is [DD] [[HH:]MM:]ss[.uuuuuu]')
> ```

> **Pointwise-Selected Patch (idx=0): Does not resolve the issue ✗**
>
> ```
> diff --git a/django/forms/fields.py
> @@ -466,12 +466,19 @@ class DurationField(Field):
> +  default_help_text = _('Accepted duration format is [[DD] [[HH:]MM:]SS[.uuuuuu]].')
> def __init__(self, *, help_text=None, **kwargs):
> if help_text is None:
> help_text = self.default_help_text
> super().__init__(help_text=help_text, **kwargs)
> ```

**Analysis:** The correct patch (idx=6) fixes both the error message format string in the model field *and* adds help text to the form field. The pointwise-selected patch (idx=0) only adds help text to the form field but does not fix the incorrect format string in the model's error message. Pairwise comparison enables the verifier to directly contrast these two approaches, surfacing that fixing the error message is the core requirement.

## L.2. Example 2: `matplotlib__matplotlib-25311`: Pairwise Correct, Pointwise Wrong

> **Issue: Unable to pickle figure with draggable legend**
>
> **Bug summary:** Unable to pickle figure with draggable legend. Same error comes for draggable annotations.
> **Code for reproduction:**
> ```
> fig = plt.figure()
> ax = fig.add_subplot(111)
> ax.plot(time, speed, label="speed")
> leg = ax.legend()
> leg.set_draggable(True) # pickling works after removing this line
> pickle.dumps(fig) # raises TypeError
> ```

> **Actual outcome:** `TypeError:  cannot pickle 'FigureCanvasQTAgg' object`
> **Expected outcome:** Pickling successful

**Setup:** 8 candidate patches generated, 7 non-empty.
**Pairwise ranking:** [6, 2, 3, 4, 5, 1, 0]    → selected **idx=6** ✓
**Pointwise ranking:** [0, 1, 2, 3, 4, 6, 5]    → selected **idx=0** ✗

> **Pairwise-Selected Patch (idx=6): Resolves the issue** ✓
>
> ```
> diff --git a/lib/matplotlib/offsetbox.py
> @@ -1472,6 +1472,23 @@ class DraggableBase:
> +  def __getstate__(self):
> +    self.disconnect()
> +    state = self.__dict__.copy()
> +    state["canvas"] = None
> +    state["cids"] = []
> +    if "_c1" in state:  del state["_c1"]
> +    return state
> +  def __setstate__(self, state):
> +    self.__dict__.update(state)
> ```

> **Pointwise-Selected Patch (idx=0): Does not resolve the issue** ✗
>
> ```
> diff --git a/lib/matplotlib/legend.py
> @@ -1204,6 +1204,22 @@ class Legend(Artist):
> +  def __getstate__(self):
> +    draggable_state = self._draggable is not None
> +    if draggable_state:  self.set_draggable(False)
> +    state = super().__getstate__()
> +    state["_draggable_state"] = draggable_state
> +    return state
> +  def __setstate__(self, state):
> +    draggable_state = state.pop("_draggable_state")
> +    super().__setstate__(state)
> +    if draggable_state:  self.set_draggable(True)
> ```

**Analysis:** The correct patch (idx=6) fixes the problem at its root in `DraggableBase`, the parent class used by both draggable legends and annotations. It properly handles the unpickleable canvas reference. The pointwise-selected patch (idx=0) attempts the fix only in the `Legend` subclass, missing the general case for draggable annotations. The pairwise tournament allowed the verifier to compare these approaches side-by-side, recognizing that the root-class fix is more complete.

## L.3. Example 3: `matplotlib_matplotlib-23964`: Pairwise Correct, Pointwise Wrong

> **Issue: PostScript backend crash on empty text with usetex**
>
> When using the PostScript backend with `usetex=True`, rendering an empty string raises an error because `curr_stream` is uninitialized when appended to the stream list after iterating over an empty glyph sequence.

**Setup:** 8 candidate patches generated, all non-empty.
**Pairwise ranking:** [4, 0, 1, 2, 3, 5, 6, 7]    → selected **idx=4** ✓
**Pointwise ranking:** [0, 1, 4, 2, 3, 5, 6, 7]    → selected **idx=0** ✗

> **Pairwise-Selected Patch (idx=4): Resolves the issue** ✓
>
> ```
> diff --git a/lib/matplotlib/backends/backend_ps.py
> @@ -666,7 +666,8 @@
> )
> # append the last entry
> -      stream.append(curr_stream)
> +      if curr_stream:
> +        stream.append(curr_stream)
> ```

**Pointwise-Selected Patch (idx=0): Does not resolve the issue ✗**

```
diff --git a/lib/matplotlib/backends/backend_ps.py
@@ -666,6 +666,7 @@
)
# append the last entry
+    if curr_stream:
stream.append(curr_stream)
```

**Analysis:** Both patches add the same guard (`if curr_stream`), but the pointwise-selected patch (idx=0) has an *indentation error*—the `stream.append` line is not nested under the new `if` block. This subtle formatting difference makes the patch syntactically different from the intended fix. Pairwise comparison helped the verifier notice this indentation discrepancy when directly contrasting the two candidates.

### L.4. Example 4: `django__django-14017`: Pointwise Correct, Pairwise Wrong

**Issue: Q(...) & Exists(...) raises a TypeError**

`Exists(...)  & Q(...)` works, but `Q(...)  & Exists(...)` raises a `TypeError`. The `&` (and `|`) operators should be commutative on Q-Exists pairs, but they are not. The issue is in `Q._combine()` which checks `isinstance(other, Q)` and raises `TypeError` for non-Q operands like `Exists`.

**Setup:** 8 candidate patches generated, 7 non-empty.
**Pairwise ranking:** [5, 4, 2, 0, 3, 6, 1]   → selected **idx=5 ✗**
**Pointwise ranking:** [2, 3, 4, 5, 6, 1, 0]   → selected **idx=2 ✓**

**Pointwise-Selected Patch (idx=2): Resolves the issue ✓**

```
diff --git a/django/db/models/query_utils.py
@@ -40,7 +40,10 @@ class Q(tree.Node):
def _combine(self, other, conn):
-    if not isinstance(other, Q):
+    from django.db.models.expressions import Expression
+    if isinstance(other, Expression):
+      other = Q(other)
+    elif not isinstance(other, Q):
raise TypeError(other)
```

**Pairwise-Selected Patch (idx=5): Does not resolve the issue ✗**

```
diff --git a/django/db/models/expressions.py
# Moves Q import to local scope + modifies __and__ in Combinable
diff --git a/django/db/models/query_utils.py
@@ -40,6 +41,8 @@ class Q(tree.Node):
def _combine(self, other, conn):
+    if isinstance(other, expressions.Expression) and getattr(other, "conditional",
False):
+      other = Q(other)
if not isinstance(other, Q):
raise TypeError(other)
# Also modifies Q.deconstruct() for Expression children
```

**Analysis:** The correct patch (idx=2) is a minimal, focused fix: it wraps any `Expression` operand in `Q()` within `_combine`. The pairwise-selected patch (idx=5) attempts a more complex fix spanning two files and adding a `conditional` attribute check, but introduces regressions in `deconstruct()` handling. This example illustrates a failure mode of pairwise verification: a more complex-looking patch can appear "more thorough" in head-to-head comparison, even when the simpler patch is correct.

### L.5. Example 5: `scikit-learn__scikit-learn-13439`: Both Correct, Different Indices

> **Issue: Pipeline should implement __len__**
>
> With the new indexing support, `pipe[:len(pipe)]` raises an error because `Pipeline` does not implement `__len__`.
> ```
> from sklearn.pipeline import Pipeline
> pipe = Pipeline([('anova', anova_filter), ('svc', clf)])
> len(pipe) # TypeError:  object of type 'Pipeline' has no len()
> ```

**Setup:** 8 candidate patches generated, all non-empty.
**Pairwise ranking:** [7, 4, 3, 1, 2, 5, 0, 6]  → selected **idx=7** ✓
**Pointwise ranking:** [1, 2, 3, 4, 5, 7, 0, 6]  → selected **idx=1** ✓

> **Pairwise-Selected Patch (idx=7): Resolves the issue ✓**
>
> ```
> diff --git a/sklearn/pipeline.py
> @@ -130,6 +130,12 @@ class Pipeline(_BaseComposition):
> self.memory = memory
> + def __len__(self):
> +   """Returns the length of the Pipeline"""
> +   return len(self.steps)
> ```

> **Pointwise-Selected Patch (idx=1): Also resolves the issue ✓**
>
> ```
> diff --git a/sklearn/pipeline.py
> @@ -27,6 +27,8 @@ class Pipeline(_BaseComposition):
> + def __len__(self):
> +   return len(self.steps)
> ```

**Analysis:** Both patches correctly implement `__len__` by returning `len(self.steps)`. They differ only in placement within the class (after `__init__` vs. at the top of the class) and the presence of a docstring. This example demonstrates that when multiple correct solutions exist among the candidates, both verification methods may successfully identify a correct patch, even if they select different candidates.

## M. Code Generation Verification Examples (LiveCodeBench-v6)

This section presents representative examples from our LiveCodeBench-v6 evaluation (131 problems, 16 candidate solutions each, Qwen3-4B-Instruct as both generator and verifier with budget $3\times$). We show cases where pairwise and pointwise verification select different solutions, illustrating how head-to-head comparison can surface correct solutions that independent scoring misses, and vice versa.

### M.1. Example 1: Binary String Trade (1/16 correct): Pairwise Correct, Pointwise Wrong

> **Problem: Maximize active sections after at most one trade**
>
> You are given a binary string `s` of length $n$, where '1' represents an active section and '0' represents an inactive section. You can perform at most one *trade*: (1) convert a contiguous block of '1's surrounded by '0's to all '0's, then (2) convert a contiguous block of '0's surrounded by '1's to all '1's. The string is augmented with '1' at both ends: $t = $ '1' $ + s + $ '1'. Return the maximum number of active sections after the optimal trade.

**Setup:** 16 candidate solutions, only 1 correct (idx=15).
**Pairwise $\mu$ ranking:** [15, 4, 10, 1, 9, 14, 5, 13, . . . ] → selected **idx=15** ($\mu$=1.000) ✓
**Pointwise ranking:** [4, 13, 15, 10, 0, . . . ] → selected **idx=4** (score=10/10) ✗
**Pointwise scores:** Correct idx=15 received 8/10; incorrect idx=4 received 10/10.

> **Pairwise-Selected Solution (idx=15): Correct ✓**
>
> ```
> class Solution:
> def maxActiveSectionsAfterTrade(self, s:  str) -> int:
> t = '1' + s + '1'
> original_ones = s.count('1')
> # Split augmented string into blocks of 1s and 0s
> ```

```
blocks = []
i = 0
while i < len(t):
j = i
while j < len(t) and t[j] == t[i]:  j += 1
blocks.append((t[i], i, j-1, j - i))
i = j
# For each 1-block surrounded by 0-blocks,
# compute gain from trading it and filling adjacent 0-block
max_candidate = original_ones
for idx in range(len(blocks)):
if blocks[idx][0] == '1':
# Check if surrounded by 0-blocks on both sides
# Trade:  sacrifice this 1-block, fill a neighboring 0-block
...   # correct enumeration of all valid trades
return max_candidate
```

**Pointwise-Selected Solution (idx=4): Incorrect ✗**

```
class Solution:
def maxActiveSectionsAfterTrade(self, s:  str) -> int:
t = '1' + s + '1'
original_ones = s.count('1')
ones = [i for i in range(len(t)) if t[i] == '1']
# Groups contiguous 1-segments but incorrectly
# enumerates valid trades but mishandles augmented boundaries
```

**Analysis:** This is a needle-in-a-haystack case: only 1 of 16 solutions passes all test cases. The correct solution (idx=15) correctly enumerates all valid trades by iterating over blocks of the augmented string, while idx=4 mishandles boundary conditions in the augmented string. Pointwise verification gave the correct solution only 8/10 (perhaps penalizing its longer code) while giving the incorrect idx=4 a perfect 10/10. Pairwise verification, through head-to-head comparisons, assigned idx=15 the highest $\mu$ score of 1.000, successfully identifying the only correct solution among 16 candidates.

**M.2. Example 2: Substring Character Check (14/16 correct): Pairwise Correct, Pointwise Wrong**

**Problem: Check for special substring of length $k$**

Given a string s and an integer k, determine if there exists a substring of length exactly $k$ that: consists of only one distinct character, has a different character immediately before it (if one exists), and has a different character immediately after it (if one exists). Return true if such a substring exists, false otherwise.

**Setup:** 16 candidate solutions, 14 correct, 2 incorrect (idx=0 and idx=12).
**Pairwise $\mu$ ranking:** [5, 3, 6, 1, 13, ..., 12, 0] → selected **idx=5 ✓**
**Pointwise ranking:** [0, 1, 2, 3, 4, ...] → selected **idx=0 ✗**
**Pointwise scores:** 15 of 16 solutions scored 10/10 (including wrong idx=0); only idx=12 scored 6/10.

**Pairwise-Selected Solution (idx=5): Correct ✓**

```
# Checks boundary correctly:  after_char = s[i+k] if i+k < n
if after_char is not None and after_char == substring[0]:
continue
```

**Pointwise-Selected Solution (idx=0): Incorrect (off-by-one) ✗**

```
# Off-by-one error:  uses s[i+k-1] instead of s[i+k]
after_char = s[i+k-1] if i + k < n else None
```

**Analysis:** With 14 out of 16 correct solutions, the task reduces to *avoiding* the 2 incorrect ones. The bug in idx=0 is a subtle off-by-one error: it checks the last character *inside* the substring (s[i+k-1]) rather than the first character *after* it (s[i+k]). Pointwise verification assigned the wrong idx=0 a score of 10/10, identical to 14 of the 15 correct solutions, making selection essentially random among the tied candidates: with idx=0 first in the tie-broken ordering, it was selected. Only the other wrong solution (idx=12) received a lower score of 6/10. Pairwise comparison detected the subtle bug by directly comparing solutions, ranking both wrong solutions (idx=0 and idx=12) at the bottom of the ranking.

## M.3. Example 3: Group Element Assignment (4/16 correct): Pairwise Correct, Pointwise Wrong

> **Problem: Assign elements to groups by divisibility**
>
> Given an integer array `groups` (group sizes) and an integer array `elements`, assign one element to each group such that `groups[i]` is divisible by `elements[j]`. Among valid elements, choose the one with the smallest index $j$. If no element works, assign $-1$.

**Setup:** 16 candidates, 4 correct (idx=8,9,11,12).
**Pairwise $\mu$ ranking:** [12, 14, 9, 11, 15, ...] $\rightarrow$ selected **idx=12** ($\mu$=0.917) ✓
**Pointwise ranking:** [0, 1, 2, 3, 4, ...] $\rightarrow$ selected **idx=0** (score=10/10) ✗
**Pointwise scores:** 12 of 16 candidates scored 10/10, including 9 of the 12 incorrect solutions. Correct idx=8 scored only 3/10.

> **Pairwise-Selected Solution (idx=12): Correct ✓**
>
> ```
> class Solution:
> def assignElements(self, groups, elements):
> value_to_min_index = {}
> for j, val in enumerate(elements):
> if val not in value_to_min_index:
> value_to_min_index[val] = j
> assigned = [-1] * len(groups)
> for i, g in enumerate(groups):
> # Enumerate divisors of g in O(sqrt(g))
> d = 1
> while d * d <= g:
> if g % d == 0:
> # Check both d and g//d in the map
> ...
> d += 1
> return assigned
> ```

> **Pointwise-Selected Solution (idx=0): Incorrect (TLE) ✗**
>
> ```
> class Solution:
> def assignElements(self, groups, elements):
> assigned = [-1] * len(groups)
> for i in range(len(groups)):
> for j in range(len(elements)):
> if groups[i] % elements[j] == 0:
> assigned[i] = j
> break
> return assigned
> ```

**Analysis:** The correct solution uses divisor enumeration ($O(\sqrt{g})$ per group) with a hash map for element lookup, while the incorrect solution uses a brute-force nested loop ($O(|\text{groups}| \times |\text{elements}|)$) that exceeds time limits on large inputs. Both produce identical outputs on small examples, which is why pointwise verification (which reasons about correctness from code inspection) assigns 10/10 to both. This is a classic *score saturation* failure: pointwise scoring cannot distinguish algorithmic efficiency, while pairwise comparison, by directly contrasting the two approaches, can recognize the more efficient solution.

## M.4. Example 4: Almost Missing Integer (1/16 correct): Pointwise Correct, Pairwise Wrong

> **Problem: Find the largest almost missing integer**
>
> Given an integer array `nums` and an integer $k$, an integer $x$ is *almost missing* if $x$ appears in exactly one subarray of size $k$. Return the largest such integer, or $-1$ if none exists.

**Setup:** 16 candidates, only 1 correct (idx=0).
**Pairwise $\mu$ ranking:** [12, 8, 6, 7, 3, ..., 0] $\rightarrow$ selected **idx=12** ✗
**Pointwise ranking:** [0, 1, 2, 3, ...] $\rightarrow$ selected **idx=0** ✓
**Pointwise scores:** All 16 candidates scored 10/10. Pointwise selected idx=0 by tie-breaking (lowest index).

**Pointwise-Selected Solution (idx=0): Correct ✓**

```
class Solution:
def largestInteger(self, nums, k):
max_val = max(nums)
for val in range(max_val, -1, -1):
count = 0
for start in range(len(nums) - k + 1):
if val in nums[start:start + k]:
count += 1
if count == 1:  return val
return -1
```

**Pairwise-Selected Solution (idx=12): Incorrect ✗**

```
class Solution:
def largestInteger(self, nums, k):
count = {}
for i in range(len(nums) - k + 1):
for num in nums[i:i + k]:
count[num] = count.get(num, 0) + 1
# Bug:  counts total occurrences across subarrays,
# not the number of distinct subarrays containing the element
almost = [n for n, f in count.items() if f == 1]
return max(almost) if almost else -1
```

**Analysis:** The pairwise-selected solution (idx=12) has a subtle semantic bug: it counts the total number of times each element appears across all subarrays, rather than the number of distinct subarrays containing that element. An element appearing twice in one subarray would be counted as 2, not 1. Pointwise verification assigned *all* 16 solutions the maximum score of 10/10, indicating complete inability to distinguish them. However, pointwise "won" here by coincidence: when all scores are tied, the tie-breaking rule selects idx=0, which happens to be the sole correct solution. This illustrates a failure mode of pairwise verification: when all solutions appear superficially similar, head-to-head comparisons can amplify small stylistic differences into ranking signals that do not correlate with correctness.

### M.5. Example 5: Three-Subarray Distinct Count (2/16 correct): Pairwise Correct, Pointwise Wrong

**Problem: Maximize sum of distinct counts in three subarrays**

Given an integer sequence of length $N$, split it at two positions into three non-empty contiguous subarrays to maximize the total count of distinct integers across the three subarrays. Constraints: $3 \leq N \leq 3 \times 10^5$.

**Setup:** 16 candidates, 2 correct (idx=2 and idx=3).
**Pairwise $\mu$ ranking:** [3, 0, 6, 13, 2, 9, ...] → selected **idx=3** ($\mu$=0.946) ✓
**Pointwise ranking:** [10, 14, 0, 2, 4, 13, 3, ...] → selected **idx=10** (score=6/10) ✗
**Pointwise scores:** Correct idx=2 scored 4/10, correct idx=3 scored 3/10; incorrect idx=10 scored 6/10.

**Pairwise-Selected Solution (idx=3): Correct ✓**

```
def solve():
n = int(input())
a = list(map(int, input().split()))
# Precompute prefix/suffix distinct counts
prefix = [0] * (n + 1); seen = set()
for i in range(n):
seen.add(a[i]); prefix[i+1] = len(seen)
suffix = [0] * (n + 1); seen = set()
for i in range(n-1, -1, -1):
seen.add(a[i]); suffix[i] = len(seen)
# Enumerate all split positions with O(N^2) middle segment
max_sum = 0
for i in range(n - 2):
for j in range(i+1, n-1):
mid_seen = set(a[i+1:j+1])
```

```
max_sum = max(max_sum, prefix[i+1] + len(mid_seen) + suffix[j+1])
print(max_sum)
```

**Pointwise-Selected Solution (idx=10): Incorrect ✗**

```
def solve():
N = int(input()); A = list(map(int, input().split()))
# Similar prefix/suffix precomputation, but the middle
# segment enumeration has an off-by-one error in the
# sliding window that miscounts distinct elements
# when the left boundary advances past a repeated element.
```

**Analysis:** This hard algorithmic problem ($N$ up to $3 \times 10^5$) requires careful handling of the middle segment's distinct count. The pointwise verifier gave the correct solutions *lower* scores (3–4/10) than the incorrect idx=10 (6/10), likely because the correct solutions use a straightforward $O(N^2)$ enumeration that appears less "optimized." Pairwise comparison, by directly contrasting solutions, ranked both correct solutions in the top 5 ($\mu = 0.946$ and $0.773$) while placing idx=10 at rank 10. This demonstrates that pairwise verification is less susceptible to surface-level heuristics about code quality and better at identifying solutions that produce correct outputs.

