# OpenReview forum: "V1: Unifying Generation and Self-Verification for Parallel Reasoners"
_ICML.cc/2026/Conference — ICML 2026 regular_

### Official Review · Reviewer_M41s · 2026-02-25

**Soundness:** 2
**Presentation:** 2
**Significance:** 2
**Originality:** 3
**Overall Recommendation:** 4
**Confidence:** 3

**Summary:**

This paper addresses the verification bottleneck in test-time scaling for LLM reasoning. Leveraging the insight that models are substantially stronger at pairwise self-verification than pointwise self-verification, the authors introduce $V_1$, a framework unifying generation and pairwise self-verification. It proposes $V_1$-Infer, an uncertainty-guided tournament algorithm for efficient candidate ranking, and $V_1$-PairRL, a unified reinforcement learning method that co-trains a single model as both generator and verifier. Experiments demonstrate that $V_1$ significantly outperforms pointwise baselines and self-aggregation methods, improving Pass@1 by up to 10% with superior efficiency.

**Compliance With Llm Reviewing Policy:**

Affirmed.

**Final Justification:**

My concerns have been adequately addressed by the authors rebuttal response and I've updated my score accordingly.

**Key Questions For Authors:**

1.Typo in Line 115：“As shown in Figure 1 (Left), the Pass@N score, representing the ...", it should be "Figure 1 (right)".
2.Why are the results for Qwen-4B-Thinking on the AIME and HMMT benchmarks not provided?

**Limitations:**

See weakness.

**Strengths And Weaknesses:**

# Strength
**Effectiveness of Pairwise Verification and Unified Co-Training**

This paper propose a pairwise verification framework designed for parallel reasonning ($V_1$-Infer), which achieves substantial performance improvements, boosting Pass@1 by up to 10% on code generation and math benchmarks compared to pointwise baselines.Furthermore, the introduction of the $V_1$-PairRL framework co-evolves the generation capability and verification capability of LLM which further improves the model performance.

# Weakness
**Lack of Statistical Significance in Base Model Experiments**

As explicitly stated in Appendix C (Line 711), "All inference experiments for base models (Section 4) are run once". However, the inference process involves significant stochasticity. Given this high degree of randomness, a single experimental run is prone to high variance, and the reported gains could partially result from lucky sampling outliers rather than the algorithmic superiority of $V_1$-Infer. To establish statistical significance, the authors should report results averaged over multiple runs with standard deviations.

---

> ### Author Rebuttal · Authors · 2026-03-31
>
> We sincerely thank the reviewer for their constructive feedback and for recognizing the effectiveness of V1's pairwise verification framework and the performance improvements achieved through our unified co-training approach. We address each concern below with substantial new experimental evidence.
>
> ---
>
> > **W1. Statistical Significance in Base Model Experiments**
>
> We thank the reviewer for this suggestion and agree that multi-seed evaluation strengthens experimental rigor. Our initial submission prioritized **cross-setting robustness** (demonstrating consistent gains across 4 models × 5 benchmarks). Our rebuttal extends this by showing improvements on 3 additional benchmarks, HealthBench (main, hard), SWE-Bench-Lite showing strong gains.
> Additionally, note that all inference results of V1-PairRL models and baselines, using V1-infer and other test-time scaling techniques, were run using 3 seeds in our original submission.
>
> We now provide results averaged over **3 random seeds** for model combinations (for off-the-shelf models), for both math and code benchmarks, using N=16 base generations.
>
> **gpt-oss-120b - AIME 2025**
>
> |Method|Mean|Std|
> |-|-|-|
> |Pass@1|76.4|1.00|
> |Pointwise|85.6|5.10|
> |RSA K=4 T=9|88.9|1.91|
> |V1-Infer (Ours)|**92.2**|1.91|
>
> **gpt-oss-120b - LiveCodeBench v6**
>
> |Method|Mean|Std|
> |-|-|-|
> |Pass@1|68.8|0.46|
> |Pointwise|74.6|1.17|
> |RSA K=4 T=9|76.1|0.87|
> |V1-Infer (Ours)|**79.4**|0.00|
>
> **qwen3-4b - LiveCodeBench v6**
>
> |Method|Mean|Std|
> |-|-|-|
> |Pass@1|34.7|1.36|
> |Pointwise|37.2|2.91|
> |RSA K=4 T=9|38.7|0.87|
> |V1-Infer (Ours)|**43.2**|0.47|
>
> **qwen3-4b - AIME 2025**
>
> |Method|Mean|Std|
> |-|-|-|
> |Pass@1|47.7|1.05|
> |Pointwise|54.5|6.93|
> |RSA K=4 T=9|**70.0**|0.00|
> |V1-Infer (Ours)|62.2|1.91|
>
> **gpt-oss-20b - AIME 2025**
>
> |Method|Mean|Std|
> |-|-|-|
> |Pass@1|65.8|0.65|
> |Pointwise|85.6|1.96|
> |RSA K=4 T=9|85.6|1.96|
> |V1-Infer (Ours)|**87.8**|1.91|
>
> **gpt-oss-20b - LiveCodeBench v6**
>
> |Method|Mean|Std|
> |-|-|-|
> |Pass@1|60.3|0.85|
> |Pointwise|72.2|3.77|
> |RSA K=4 T=9|69.0|2.71|
> |V1-Infer (Ours)|**75.8**|0.87|
>
> **Qwen3-4B-Thinking - LiveCodeBench v6**
>
> |Method|Mean|Std|
> |-|-|-|
> |Pass@1|52.8|0.42|
> |Pointwise|58.4|0.57|
> |RSA K=4 T=9|52.7|0.0|
> |V1-Infer (Ours)|**63.4**|2.19|
>
> For Qwen3-4B-Thinking results on AIME and HMMT, see Q2 below.
>
> **Key observations:**
>
> - **V1-Infer consistently outperforms** both Pass@1 and Pointwise across all seeds and settings, and outperforms or matches RSA on 8 out of 9 settings.
> - **V1-Infer exhibits lower variance** than Pointwise in 7 out of 8 settings (e.g., Std 0.87 vs 3.77 on gpt-oss-20b LCBv6; Std 1.91 vs 5.10 on gpt-oss-120b AIME). This indicates that V1-Infer provides a more stable signal for self-verification.
> - Combined with the cross-setting consistency across 4 models and 5 benchmarks in our original submission, these multi-seed results confirm that the gains are robust and not attributable to sampling variance.
>
> We note that RSA requires up to 2x more output tokens and 10x more input tokens compared with V1-Infer. We discuss this trade-off further in our response to Reviewer m19t (Q1).
>
> ---
>
> > **Q1. Typo in Line 115**
>
> Thank you for catching this. We will fix it.
>
> ---
>
> > **Q2. Missing Qwen-4B-Thinking Results on AIME and HMMT**
>
> We appreciate this question. We initially omitted Qwen-Thinking results on math benchmarks due to its high evaluation cost (32K generation length). We have now completed these experiments with 3 seeds:
>
> **Qwen3-4B-Thinking - AIME 2025**
>
> |Method|Mean|Std|
> |-|-|-|
> |Pass@1|70.2|1.46|
> |Pointwise|67.8|1.91|
> |RSA K=4 T=9|86.2|0.35|
> |V1-Infer (Ours)|**86.5**|3.87|
>
> **Qwen3-4B-Thinking - HMMT Feb 2025**
>
> |Method|Mean|Std|
> |-|-|-|
> |Pass@1|49.8|1.11|
> |Pointwise|56.7|6.65|
> |RSA (taken from Venkatraman et al.)|62.4|-|
> |V1-Infer (Ours)|**67.8**|1.91|
>
> V1-Infer outperforms pointwise by +17.7pp on AIME and +11.1pp on HMMT. On HMMT, V1-Infer (67.8) also outperforms RSA (62.6). Note that RSA requires up to 2.7x extra cost compared to V1-Infer (see Reviewer m19t, Q1).
>
> ---
>
> > **Additional Evidence: Generalization Beyond Verifiable Domains**
>
> Since several reviewers raised concerns about generalization, we briefly note that we have also evaluated V1-Infer on **non-verifiable, open-domain medical QA** (HealthBench) and trained V1-PairRL with **rubric-based rewards** on a medical domain. V1-Infer shows +3-27% gains over pointwise across 3 models on HealthBench, and V1-PairRL with rubric rewards outperforms both pointwise co-training and baseline RL. Full details are provided in our response to Reviewer o4ZX (W2).
>
> ---
>
> Please let us know if you have any other questions and we hope we were able to resolve your concerns about statistical significance. Since that was the only weakness, if you feel your concerns have been addressed, we would like to very politely request an update to your score.

---

> > ### Author Rebuttal · Reviewer_M41s · 2026-04-01
> >
> > Thanks for the comprehensive response. My concerns have been adequately addressed and I've updated my score accordingly.

---

> > > ### Author Response · Authors · 2026-04-05
> > >
> > > Dear Reviewer M41s,
> > >
> > > Thank you for taking the time to carefully review our rebuttal and for confirming that your concerns have been fully resolved. We truly appreciate the constructive feedback on statistical significance, which motivated us to run multi-seed evaluations that we believe meaningfully strengthen the paper.
> > >
> > > During the rebuttal, for answering questions for all reviewers, we obtained several new results that strengthened the paper beyond the original submission by adding supporting evidence to our original results. We briefly summarize them here for your convenience:
> > >
> > > > **Summary of Rebuttal**
> > >
> > > - **Multi-seed evaluations** (3 seeds) across math and code model-benchmark combinations, showing V1-Infer consistently outperforms baselines with lower variance than pointwise in 7 out of 8 settings.
> > > - **Qwen3-4B-Thinking on AIME and HMMT** (addressing Q2): +17.7pp over pointwise on AIME, +11.1pp on HMMT, outperforming RSA on HMMT.
> > > - **V1-PairRL on Math-RL** (Qwen3-4B-Base, GRPO): +5.7pp on MATH-Hard, 2x on AIME over pointwise, with +215% judge reward improvement during training showing that pairwise self-verification capability is a trainable capability.
> > > - **V1-PairRL on non-verifiable medical QA** using rubric-based rewards (HealthBench, RaR-Medicine), showing gains over both pointwise co-training and baseline RL.
> > > - **V1-Infer on HealthBench** across 3 models with +3-27% gains over pointwise, demonstrating generalization to non-verifiable, open-domain settings.
> > > - **Comparisons against 4 external verifiers** (AceMath-7B-RM, Skywork-o1-PRM, Qwen-Math-PRM, ThinkPRM-7B), all of which underperform V1-Infer.
> > > - **Token-level efficiency analysis** showing V1-Infer achieves higher accuracy than RSA at 2-2.7x lower cost.
> > > - **SWE-Bench-Lite evaluation**: V1-Infer achieves 33.3% vs 28.3% pointwise (+5pp) on full agent trajectories.
> > >
> > > We believe these results collectively demonstrate the generality, efficiency, and robustness of V1 across domains, models, and evaluation settings, and we hope you could take these under consideration for your Final Justification and Overall Recommendation for our work.
> > >
> > > Thank you again for your support of our paper and valuable feedback.

---

### Official Review · Reviewer_m19t · 2026-03-14

**Soundness:** 2
**Presentation:** 3
**Significance:** 3
**Originality:** 3
**Overall Recommendation:** 5
**Confidence:** 3

**Summary:**

This paper introduces V1, a framework to improve the performance of TTS by unifying solution generation and self-verification. This paper first argues that pointwise verification (one model scores solutions in isolation) suffers from poor calibration, and self-aggregation often lead to diversity collapse by discarding correct outlier solutions. To address these, this paper propose V1-infer, an uncertainty-guided algorithm that uses a tournament-based ranking system to allocate verification compute to candidate pairs whose relative correctness is most uncertain. Additionally, they further introduce the V1-pairrl, a RL framework that jointly trains a single model as both a generator and a pairwise verifier, ensuring the verifier adapts as the generator's capabilities evolve. Evaluation is performed on code generation (LCB and CodeContests) and math reasoning (AIME and HMMT) using GPT-20B and Qwen3-4B, the proposed method improves Pass@1 and pointwise self-verification.

**Compliance With Llm Reviewing Policy:**

Affirmed.

**Final Justification:**

I think the authors have addressed most of my concerns. I will maintain my positive score.

**Key Questions For Authors:**

I only have a few questions for the authors:
1.  Why do you only compare RSA on LCBv6? What about other benchmarks?
2. For FIgure 7 at appendix, for Qwen3-4B-Thinking, why using budgets 1x is better than 2x on LCBv5 and LCBv6, seems a bit counterintuitive.

**Limitations:**

Yes

**Strengths And Weaknesses:**

Strengths:
1. Presentation: This paper is well-organized and easy to follow.
2. Motivation: I think this paper is well-motivated. They first identify calibration collapse of pointwise verification and diversity collapse under self-aggregation. Then they propose V1-infer to shift from pointwise to pairwise self-verification for parallel reasoning. V1-PairRL is further introduced to train a unified framework that co-evolves a single model as both generator and verifier, including concrete anti-collusion measures.
3. Significance: This paper focuses on the key bottleneck of TTS by using self verification without external verifier and co-training of pairwise verification. It seems to be a promising direction for building self verifying models that better exploit parallel reasoning at scale.
4. Experiments: The authors conducted experiments on two domains: code and math by using 5 benchmarks, and used 4 open-sourced LLMs (2 results are reported in the Appendix: GPT-OSS 120B and Qwen3-4B-Thinking) from 2 families. For TTS, they compare with pass@1, pointwise self-verification and recursive self-aggregation (only on LCBv6). For training, they compare with pointRL and vanilla RL baseline.

Weaknesses:
My main concerns are mostly about the experimental results, including:
1. The baselines used for evaluation are relatively limited (only two and RSA on LCBv6). It would be interesting to see the performance comparisons with strong TTS methods that leverage external verifiers or step-level PRMs and rank-aggregation approaches.
2. The metric of efficiency is using the number of verification calls rather than tokens or wall-clock time. Pairwise method includes two full solutions and could be more token-expensive than baselines.
3. V1-pairrl depends on verifiable domains to construct verifier rewards and to enforce the at least one correct pairing constraint. Generalization to non-verifiable settings is unclear.
4. Related work: would be beneficial to discuss existing works on pairwise ranking (e.g. [1][2]), especially [2] also uses uncertainty-guided pairwise-preference search.

References:

[1]Large language models are effective text rankers with pairwise ranking prompting. Findings of ACL 2024.

[2]Aligning with Human Judgement: The Role of Pairwise Preference in Large Language Model Evaluators. COLM 2024.

---

> ### Author Rebuttal · Authors · 2026-03-31
>
> We thank the reviewer for the positive assessment and for recognizing the significance of our work. We address each point below.
>
> ---
> > **W1. Limited Baselines: External Verifiers and PRMs**
>
> Our work focuses on *self-verification* hence we compared against verifier-free test-time scaling strategies. Following your suggestion, we compare against 4 external reward models (AceMath-7B-RM, Skywork-o1-PRM, Qwen-Math-PRM, ThinkPRM-7B) using best-of-N selection across 3 models. On AIME 2025 (N=16):
>
> |Method|Type|gptoss-120b|gptoss-20b|qwen3-4b|
> |-|-|-|-|-|
> |Pass@1|random|76.0|65.2|49.2|
> |BoN AceMath-7B-RM|Ext. ORM|60.0|63.3|43.3|
> |BoN Skywork-o1-PRM|Ext. PRM|73.3|60.0|36.7|
> |BoN Qwen-Math-PRM|Ext. PRM|76.7|56.7|56.7|
> |BoN ThinkPRM-7B|Ext. Gen. PRM|73.3|70.0|40.0|
> |Pointwise|Self-verify|85.6+-5.10|85.6+-1.96|54.5+-6.93|
> |V1-Infer (Ours)|Self-verify|**92.2**+-1.91|**87.8**+-1.91|**62.2**+-1.91|
>
> LCBv6 results show similar trends (link for [full comparison graph](https://image2url.com/r2/default/images/1774956963077-d7d1a830-e840-4c03-b5e3-4e068cc7e44c.png)). **External ORMs/PRMs generally underperform self-verification.** We hypothesize and qualitatively observe a **distribution mismatch**: these RMs were trained on older qwen-family outputs, and we see cases where correct reasoning steps are marked as wrong. Qwen-Math-PRM helps on qwen3-4b where the training distribution may partially overlap, but underperforms on other models. V1-Infer consistently outperforms all external verifiers across both benchmarks and all models. Additional verifier-free baselines (Self-Refine, rejection sampling) underperform RSA (Venkatraman et al., 2025), which V1-Infer already outperforms (see Q1).
>
> ---
> > **W2. Efficiency Metric Uses Verification Calls, Not Tokens ...**
>
> **Input (prefill) tokens are ~4-8x cheaper than output (decode) tokens** in standard API pricing (GPT-4o: 4x, Gemini 2.5 Pro: 8x). We define *effective tokens* = Output + 0.20 x Input as a cost proxy. For gptoss-120b with N=16 (eff. tokens are dataset totals, averaged over 3 seeds):
>
> |Method|AIME Acc|AIME Eff. Tokens|LCBv6 Acc|LCBv6 Eff. Tokens|
> |-|-|-|-|-|
> |Pointwise N=32 (64 calls)|83.3|7.7M|75.6|27.5M|
> |RSA K=4 T=9|88.9|17.0M|76.1|87.0M|
> |**V1-Infer N=16 3N (64 calls)**|**92.2**|**6.8M**|**79.4**|**31.6M**|
>
> V1-Infer wins on accuracy while being **cheaper than pointwise N=32** on AIME (0.88x) and comparable on LCBv6 (1.15x). **RSA is 2.0-2.7x more expensive** while achieving lower accuracy. For gptoss-20b, V1-Infer similarly outperforms pointwise N=32 (87.8 vs 86.7 on AIME at 0.80x cost; 75.8 vs 71.0 on LCBv6) and RSA (which is 2.0-2.4x more expensive). For qwen3-4b, V1-Infer outperforms pointwise N=32 at near-identical cost (1.02x on both AIME and LCBv6).
>
> ---
> > **W3. Depends on Verifiable Domains**
>
> V1-PairRL is applicable to **any RL method and domain**. We train on a non-verifiable medical QA domain using rubric-based rewards following (Gunjal et al., Rubrics as Rewards, 2025). Qwen3-4B-Instruct trained on RaR-Medicine (17,926 QA) with GRPO, 100 steps. The policy pairwise self-verifies *without access to dataset rubrics*. see link for [Experimental Setup](https://bit.ly/image1774958758314).
>
> |Benchmark|Method|Pass@1 (response score)|
> |-|-|-|
> |HealthBench|V1-PointRL|0.237|
> ||**V1-PairRL (Ours)**|**0.275**|
> |RaR-Medicine Test|V1-PointRL|0.481|
> ||**V1-PairRL (Ours)**|**0.506**|
>
> **V1-PairRL (0.275) > Pointwise (0.237)** on HealthBench. Notably, V1-PairRL also scores above baseline GRPO with rubric rewards (0.256 pass@1 without self-verification). On RaR-Medicine, V1-PairRL outperforms pointwise (0.506 vs 0.481). We also evaluate V1-Infer on HealthBench at inference time across 3 models (+3-27% gains); full results in our response to Reviewer o4ZX (W2).
>
> ---
> > **W4. Missing Related Work on Pairwise Ranking**
>
> Thank you. We will discuss both works in detail. Our Swiss refinement is related to [2]'s uncertainty-guided search, but we focus on *self*-verification in parallel reasoning with co-training for test-time scaling, while [2] studies external LLM evaluators for alignment.
>
> ---
> > **Q1. Comparing RSA beyond LCBv6**
>
> Multi-seed RSA comparisons across 9 model-benchmark settings are in Reviewer M41s's response (W1) showing V1-Infer improves over RSA on **8 out of 9 settings**. Additionally, we also evaluate RSA on non-verifiable domains (HealthBench) where it shows severe diversity collapse: on HealthBench Hard with gptoss-120b, pass@1 drops from 0.303 to 0.035 within 3 loops and stays flat.
>
> ---
> > **Q2. Budget 1x > 2x for Qwen3-Thinking**
>
> This appeared in the N=8 setting (Figure 7) where the small candidate pool increases variance. With N=16, we see **monotonic improvements**:
> - AIME: 80.0 (1x) to 83.3 (2x) to 83.3 (3x)
> - HMMT: 66.7 (1x) to 70.0 (2x) to 70.0 (3x)
> - LCBv6: 54.2 (1x) to 56.5 (2x) to 56.5 (3x)
>
> With sufficient candidate diversity, V1-Infer scales monotonically with verification budget.

---

> > ### Author Rebuttal · Reviewer_m19t · 2026-04-01
> >
> > I think the authors have addressed most of my concerns. I will maintain my positive score.

---

> > > ### Author Response · Authors · 2026-04-05
> > >
> > > Dear Reviewer m19t,
> > >
> > > Thank you again for your valuable and constructive feedback, suggestions, and the time spent reviewing our work and rebuttal in detail. We believe your questions and suggestions helped us improve our paper and showcase the generality of our algorithms to multiple domains (such as non-verifiable domains), as well as make the efficiency and performance gains over prior methods clearer.
> > >
> > > Thank you for your support of our work.

---

### Official Review · Reviewer_o4ZX · 2026-03-14

**Soundness:** 3
**Presentation:** 3
**Significance:** 2
**Originality:** 1
**Overall Recommendation:** 4
**Confidence:** 4

**Summary:**

This paper addresses a fundamental bottleneck in test-time scaling for LLMs: when sampling multiple parallel solutions, how do you reliably identify the correct one? The authors argue that standard pointwise (scalar) self-verification is poorly calibrated, and that pairwise comparison is a more natural and effective primitive. They introduce V1, comprising two components: V1-Infer (an inference-time tournament algorithm) and V1-PairRL (a unified RL training framework).

**Compliance With Llm Reviewing Policy:**

Affirmed.

**Final Justification:**

After considering the authors response to my questions. I'm happy to update my score as my concerns have been positively addressed.

**Key Questions For Authors:**

Please refer to weaknesses 1 and 2.

**Limitations:**

Please refer to weaknesses.

**Strengths And Weaknesses:**

- The core insight is well-motivated. The authors draw on established statistical theory — Bradley-Terry models, active learning principles — to explain why pointwise verification fails (calibration collapse) and why pairwise ranking should be preferred.

- The two failure modes they identify and characterize — calibration collapse in pointwise scoring and diversity collapse in self-aggregation methods like RSA — are clearly demonstrated and provide a coherent narrative that motivates the proposed approach.

- V1-PairRL's treatment of reward hacking is particularly noteworthy. The authors identify two specific collapse modes (safe bet collapse and the empty solution loop) and address each with targeted design choices: a sparsity threshold on verification rewards and a strict pairing strategy requiring at least one correct solution per training pair.

- The central idea — that it is easier and more reliable to compare two solutions against each other than to score each one in isolation — is immediately graspable and requires little conceptual overhead to appreciate or implement. This simplicity is a genuine virtue: the algorithm introduces no exotic machinery, relies only on standard prompting of the underlying model, and can be reimplemented from the paper's pseudocode and prompt templates with minimal effort.

***Weaknesses***

- V1-PairRL is evaluated only on code generation (following the DeepCoder recipe). Whether the co-training benefits transfer to math reasoning — where the paper's inference results are also strong — is an open question. Given that V1-Infer delivers strong results on AIME and HMMT, this omission is conspicuous and warrants a clearer justification in the authors' response.

- Perhaps the most significant open question left unaddressed by this work is whether pairwise self-verification retains its advantages beyond tasks with objectively verifiable ground truth. Code generation and mathematical reasoning are, by design, among the most tractable domains for any verification approach: correctness is well-defined, test cases can arbitrate disputes, and the space of valid answers is tightly constrained. The paper's framing implicitly assumes that models can serve as reliable comparative judges, but this assumption becomes considerably less stable in semantically richer reasoning tasks such as open-domain question answering, commonsense reasoning, long-form scientific analysis, or hypothesis generation — domains where the boundary between correct and incorrect is inherently blurry. In these settings, pairwise verification may simply substitute one form of bias for another: rather than the calibration collapse observed in pointwise scoring, the model may instead exhibit preference for superficially appealing responses — those that are more fluent, more confidently worded, or longer — rather than those that are genuinely more correct.

- A recurring concern with this work, despite its empirical merits, is that the individual components of V1 are largely assemblies of well-established ideas rather than genuinely novel theoretical contributions. Pairwise comparison as a more reliable signal than absolute scoring is a long-standing observation in the preference learning and reward modeling literature. The contribution of this paper lies primarily in their careful combination and empirical validation in the specific context of parallel LLM reasoning (partly math and code).

---

> ### Author Rebuttal · Authors · 2026-03-31
>
> We thank the reviewer for the thoughtful review and for recognizing our well-motivated framing, the treatment of reward hacking, and the simplicity of our approach.
>
> ---
>
> > **W1. V1-PairRL Evaluated Only on Code Generation**
>
> Since V1-PairRL applies to any RL algorithm and domain, as per the reviewers' suggestions, we conduct further experiments for support, on two new domains:
>
> 1. **Math-RL:** Qwen3-4B-Base trained on Polaris-53K for ~300 steps. N=16, self-verification based test-time scaling:
>
> |Method|MATH-Hard|AIME 2025|Verif. Acc (MATH-Hard)|
> |-|-|-|-|
> |V1-PointRL|62.5|10.0|63.9|
> |**V1-PairRL (Ours)**|**68.2**|**20.0**|**71.5**|
>
> V1-PairRL achieves **68.2** vs 62.5 on MATH-Hard (+5.7pp) with higher verification accuracy (71.5 vs 63.9). On AIME, pointwise *hurts* (10.0 vs 15.6 baseline), while pairwise improves it to 20.0. V1-PairRL also improves base generation quality. On MATH-Hard pass@1 for base generations from V1-PairRL 62.5 vs 58.7 (V1-PointRL) and 56.0 (Baseline GRPO).
>
> 2. **Rubric-based RL for non-verifiable domains:**
> Qwen3-4B-Instruct trained on RaR-Medicine (17,926 medical QA) with GRPO, 100 steps. The policy pairwise self-verifies *without access to dataset rubrics*. See [Table with Experimental Setup details](https://bit.ly/image1774958758314).
>
> |Benchmark|Method|Pass@1 w/ self-verification|
> |-|-|-|
> |HealthBench Main|V1-PointRL|0.237|
> ||**V1-PairRL (Ours)**|**0.275**|
> |RaR-Medicine Test|V1-PointRL|0.481|
> ||**V1-PairRL (Ours)**|**0.506**|
>
> **V1-PairRL (0.275) > Pointwise (0.237)** on HealthBench, also exceeding baseline GRPO (0.256 pass@1). On RaR-Medicine test set, V1-PairRL outperforms pointwise (0.506 vs 0.481).
>
>
> Combined with Code-RL in the paper, V1-PairRL shows consistent gains across **three domains**: code, math, and non-verifiable medical QA.
>
> ---
>
> > **W2. V1-Infer's generalization to non verifiable settings ... may instead exhibit preference for superficially appealing responses**
>
> Since V1-Infer and V1-PairRL do not assume access to ground truth at inference time, the approaches are applicable to any domain. We evaluate on HealthBench and HealthBench-Hard, high quality, and non-verifiable medical QA (open-domain) benchmarks.
>
> **V1-Infer on HealthBench (N=16, GPT-5-nano grader, 200 random examples per benchmark):**
>
> |Benchmark|Model|V1-Infer|Pointwise|Gain|
> |-|-|-|-|-|
> |HealthBench Hard|gpt-oss-20b|**0.219**|0.172|**+27%**|
> ||gpt-oss-120b|**0.356**|0.324|**+10%**|
> ||qwen3-4B-instruct|**0.170**|0.135|**+26%**|
> |HealthBench Main|gpt-oss-20b|**0.495**|0.474|**+4%**|
> ||gpt-oss-120b|**0.586**|0.571|**+3%**|
> ||qwen3-4B-instruct|**0.420**|0.391|**+7%**|
>
> V1-Infer consistently outperforms pointwise across all 3 models (+3-27%). Harder problems benefit more (larger gains on Hard vs Main). **RSA shows diversity collapse here too:** on HealthBench Hard with gptoss-120b, pass@1 drops from 0.303 to 0.035 within 3 loops.
>
> **Does pairwise prefer superficially appealing responses?** We measure Spearman correlation between verification scores and response length (LOWER is better):
>
> |Benchmark|Model|V1-Infer score vs Length|Pointwise score vs Length|
> |-|-|-|-|
> |HealthBench Hard|gpt-oss-20b|**0.330**|0.373|
> ||gpt-oss-120b|**0.189**|0.236|
> ||Qwen3-4B|**0.086**|0.088|
> |HealthBench Main|gpt-oss-20b|**0.218**|0.280|
> ||gpt-oss-120b|**0.146**|0.182|
> ||Qwen3-4B|**0.086**|0.118|
>
> **Pointwise is more length-biased than V1-Infer across all 6 combinations.** Pairwise is more calibrated even in non-verifiable settings.
>
> We also train V1-PairRL on this non-verifiable domain with rubric-based rewards; see W1 above and our response to Reviewer yr9T (W2) for further details.
>
> ---
>
> > **W3. On Novelty**
>
> We acknowledge pairwise comparison is established in RLHF (Section 2) with external reward models. However, our main focus is on studying and improving **self-verification**. Prior pairwise work (PairRM, AggLM) trains *separate* verifiers on curated data, while we study whether a model can verify **its own outputs**. Recent work (Lu et al., 2025; "When Does Verification Pay Off?") concludes LLMs cannot reliably self-verify, and our work is the first to substantially improve self-verification capabilities using 1) a novel inference algorithm (**V1-Infer**) with uncertainty-based budget allocation that is up to **2.7x cheaper** and more performant than RSA (see Reviewer m19t's response for cost analysis), and 2) a novel co-evolving RL recipe (**V1-PairRL**) with anti-reward-hacking mechanisms (as the reviewer noted in Strength S3, the collapse mode identification and mitigations are noteworthy) that improves pairwise self-verification across **code, math, and non-verifiable domains**. Together, these form a **unified system** that identifies *why* self-verification fails (calibration collapse, diversity collapse) and addresses both inference and training, validated across 4 models and 8 benchmarks.

---

> > ### Author Rebuttal · Reviewer_o4ZX · 2026-04-06
> >
> > Thank you for addressing my questions and for the additional experiments. My concerns have now been addressed. I will update  my score.

---

> > > ### Author Response · Authors · 2026-04-07
> > >
> > > Dear Reviewer o4ZX,
> > >
> > > Thank you again for your valuable and constructive feedback, suggestions, and the time spent reviewing our work and rebuttal in detail. We believe your questions and suggestions helped us improve our paper, showcase the generality of our algorithms across multiple domains, including math and non-verifiable domains.
> > >
> > > During the rebuttal, for answering questions for all reviewers, we obtained several new results that strengthened the paper beyond the original submission by adding supporting evidence to our original results.
> > >
> > > We briefly summarize them here for your convenience:
> > >
> > > > **Summary of Rebuttal**
> > >
> > > - **Multi-seed evaluations** (3 seeds) across math and code model-benchmark combinations, showing V1-Infer consistently outperforms baselines with lower variance than pointwise in 7 out of 8 settings.
> > > - **Qwen3-4B-Thinking on AIME and HMMT** (addressing Q2): +17.7pp over pointwise on AIME, +11.1pp on HMMT, outperforming RSA on HMMT.
> > > - **V1-PairRL on Math-RL** (Qwen3-4B-Base, GRPO): +5.7pp on MATH-Hard, 2x on AIME over pointwise, with +215% judge reward improvement during training showing that pairwise self-verification capability is a trainable capability.
> > > - **V1-PairRL on non-verifiable medical QA** using rubric-based rewards (HealthBench, RaR-Medicine), showing gains over both pointwise co-training and baseline RL.
> > > - **V1-Infer on HealthBench** across 3 models with +3-27% gains over pointwise, demonstrating generalization to non-verifiable, open-domain settings.
> > > - **Comparisons against 4 external verifiers** (AceMath-7B-RM, Skywork-o1-PRM, Qwen-Math-PRM, ThinkPRM-7B), all of which underperform V1-Infer.
> > > - **Token-level efficiency analysis** showing V1-Infer achieves higher accuracy than RSA at 2-2.7x lower cost.
> > > - **SWE-Bench-Lite evaluation**: V1-Infer achieves 33.3% vs 28.3% pointwise (+5pp) on full agent trajectories.
> > >
> > > We believe these results collectively demonstrate the generality, efficiency, and robustness of V1 across domains, models, and evaluation settings.
> > > We hope you could take these under consideration for your Final Justification and Overall Recommendation for our work.
> > >
> > > Thank you again for your support of our paper and valuable feedback.

---

### Official Review · Reviewer_yr9T · 2026-03-24

**Soundness:** 2
**Presentation:** 3
**Significance:** 2
**Originality:** 2
**Overall Recommendation:** 3
**Confidence:** 4

**Summary:**

The paper proposes a framework to improve how LLMs solve difficult reasoning tasks. Instead of scoring each generated answer independently, which often fails due to poor calibrations, the authors show that models perform much better when comparing answers in pairs.
Based on this, they introduce two things. First, a tournament-style selection algorithm that compares candidate answers pairwise and focuses computation on the most uncertain comparisons. Second, a joint training method where a single model learns to both generate and evaluate answers, allowing its judgment to improve as its reasoning advances.
Overall, they show that pairwise comparison makes answer selection more reliable, the tournament method improves efficiency and accuracy on tasks like math and coding, and joint training leads to stronger reasoning than previous approaches.

**Compliance With Llm Reviewing Policy:**

Affirmed.

**Final Justification:**

My major concerns remain after the rebuttal. Details are in the response messages.

**Key Questions For Authors:**

See weaknesses

**Limitations:**

See weaknesses

**Strengths And Weaknesses:**

Strengths:

- The paper empicirally proves that comparing two answers side by side works much better than scoring answers one by one. The authors show that their pairwise ranking method increases top accuracy by a decent amout of gain on coding and math, compared to standard single answer scoring. It also performs well on hard problems, showing an accuracy jump on difficult coding tasks where the model needs the most help picking the right solution.

- Instead of comparing every single possible pair of answers, their tournament algorithm only focuses on comparing the answers that are most uncertain or have similar scores. This allows the model to match other methods while using fewer steps to check the work. It also avoids throwing away unique correct answers, which is a common flaw in methods that try to blend multiple answers.

- The writing is clearn and plots illustrate well the main ideas of the work.


Weaknesses:

- The main idea of comparing two answers side by side is not completely new, as it has been widely used before to train models using human feedback. The authors take this existing comparison method and apply it to a slightly different setup - training the model to check its own generated answers. Because it builds on past techniques without introducing new concepts, it feels like a small step forward rather than a breakthrough.

- The training method, V1 pair rl,  specifically requires tasks where we can automaticaly verify if an answer is right or wrong, such as running  code or checking a math solution. This strict requirement means the framework would struggle with more open ended tasks where there is no single exact answer. This limits the usefulness of their method to a narrow set of subjects.

- I didn't see comparisons with existing RL methods on LLMs. These should be considered as baselines too. If at least one baseline is added, the paper could be strengthened. also, the authors did show that  v1 infer improve over pointwise method under same compute budget, but still there is a lack of comparisons of existing test-time scaling methods.

- It seems to me that the success of the methods depend heavily on the model already having some underlying ability to judge pairs of answers correctly. If a problem is so difficult that the model cannot even recognize a better step when it sees one, comparing answers side by side might not produce the right solution. authors should ablate this point more in the experiment section.

---

> ### Author Rebuttal · Authors · 2026-03-31
>
> We thank the reviewer for recognizing the strengths of our approach. We address each concern below.
>
> ---
> > **W2. Restricted to Verifiable Domains**
>
> V1-PairRL (and V1-Infer) is a general approach applicable to **any RL method and domain**. To demonstrate this, we train on a non-verifiable medical QA domain using rubric-based rewards following (Gunjal et al., Rubrics as Rewards, 2025).
>
> **V1-PairRL with rubric-based RL:** Qwen3-4B-Instruct trained on RaR-Medicine (17,926 medical QA) with GRPO, 100 steps. The policy pairwise self-verifies *without access to dataset rubrics*. ([See Exp. Setup Table](https://bit.ly/image1774958758314)).
>
> Results:
>
> |Benchmark|Method|Pass@1 (response score)|
> |-|-|-|
> |HealthBench|V1-PointRL|0.237|
> ||**V1-PairRL (Ours)**|**0.275**|
> |RaR-Medicine Test|V1-PointRL|0.481|
> ||**V1-PairRL (Ours)**|**0.506**|
>
> **V1-PairRL (0.275) > Pointwise (0.237)** on HealthBench; V1-PairRL similarly outperforms pointwise on RaR-Medicine (0.506 vs 0.481).
>
> **V1-Infer on HealthBench** (N=16, GPT-5-nano grader): V1-Infer outperforms pointwise by +3-27% across 3 models on HealthBench Main and Hard ([Full Results](https://image2url.com/r2/default/images/1774947487419-25ae79f7-d307-4cb0-a28a-89163dc3912b.png)). RSA also exhibits diversity collapse on HealthBench: pass@1 drops from 0.303 to 0.035 within 3 loops.
>
> ---
> > **W3. Missing RL Baselines ... Comparisons with Test-Time Scaling Methods**
>
> V1-PairRL is *orthogonal* to the choice of RL algorithm and can be applied on top of any RL method. In our paper, we use DAPO (Section 5.3). We now show results across three domains:
>
> - **Code-RL with DAPO (paper, Section 5.4):** V1-PairRL achieves +6.5-7.3% over V1-PointRL across LCB-v5, LCB-v6, and CodeContests, and improves base Pass@1 by +2.7-8.7% over the RL baseline.
> - **Math-RL with GRPO (new):** Qwen3-4B-Base trained on Polaris-53K for 300 steps. N=16, self-verification based test-time scaling:
>
> |Method|MATH-Hard|AIME 2025|Verif. Acc (MATH-Hard)|
> |-|-|-|-|
> |V1-PointRL|62.5|10.0|63.9|
> |**V1-PairRL (Ours)**|**68.2**|**20.0**|**71.5**|
>
> V1-PairRL achieves **68.2** vs 62.5 on MATH-Hard (+5.7pp) with higher verification accuracy (71.5 vs 63.9). On AIME, pointwise verification actually *hurts* performance (10.0 vs 15.6 baseline), while pairwise improves it to 20.0. Judge reward improves from 0.19 to 0.59 (+215%) during training, confirming trainability (see W4).
>
> - **Rubric RL (non-verifiable, new):** See W2 above.
>
> **Test-time scaling comparisons:** We compare V1-Infer against 4 external verifiers (AceMath-7B-RM, Skywork-o1-PRM, Qwen-Math-PRM, ThinkPRM-7B) and RSA. External verifiers underperform self-verification due to distribution mismatch. V1-Infer outperforms all on AIME and LCBv6, and beats RSA at 2-3x lower token cost. Full results in our response to Reviewer m19t (W1).
>
> ---
> > **W4. Dependence on Model's Inherent Pairwise Judgment Ability**
>
> Across all settings, we find that pairwise judgment is consistently better than pointwise, and importantly, this capability is trainable even for models that initially lack it or are poor at it. V1-PairRL achieves large self-verification reward increases during training, e.g., +215% on Math-RL (0.19 to 0.59) with qwen-4b-base.
>
> **Following the reviewer's suggestion we benchmark V1 on harder tasks**:
> - **SWE-Bench-Lite** (300 examples, Gemini 2.5 Flash): Base 26.3%, pointwise 28.3%, V1-Infer **33.3%** (+5pp), verifying full agent trajectories.
> - **HealthBench-Hard**: up to +27% relative gain over pointwise (see W2).
> - **LiveCodeBench by difficulty** (Fig. 4 in paper): V1-Infer provides the largest gains on hard problems (+23.7%), where Pass@1 is lowest.
>
> We agree some pairwise judgment capability is required; our paper specifically claims that existing models are already quite capable at pairwise self-verification, which leads to consistent improvements over the strongest test-time scaling methods like RSA across all tested settings.
>
> ---
> > **W1. Limited Novelty: Pairwise Comparison Is Not New**
>
> We acknowledge pairwise comparison is established in RLHF (Section 2) with external reward models. However, our main focus is on studying and improving *self-verification*. Recent work (Lu et al., 2025; "When Does Verification Pay Off?") concludes LLMs cannot reliably self-verify. Our core contribution is showing **pairwise self-verification overcomes this**, bridging pass@1 to pass@N where pointwise fails, validated across 4 models and 8 benchmarks. This is distinct from prior work (PairRM, AggLM) using *separate* external verifiers. For this we develop **V1-Infer**, a novel algorithm applying uncertainty based budget allocation (+3.8% performance over random, O(N) comparisons, and up-to 2.7x less expensive than RSA), and we propose for the first time, a novel **V1-PairRL** methodology, co-evolving solution generation and pairwise self-verification capability for which we developed new reward hacking mitigation schemes. See Reviewer o4ZX response for further discussion.

---

> > ### Author Rebuttal · Reviewer_yr9T · 2026-04-04
> >
> > I appreciate the additional experiments, particularly those extending the method to rubric-based and non-verifiable domains. That said, my overall assessment remains largely unchanged. While the results look solid and the engineering contributions are clear, I still find the core idea, leveraging pairwise comparison for evaluation and training, to be a relatively incremental extension of existing paradigms rather than "a novel algorithm", and I don't think "reaching a SOTA performance" justifies a solid contribution to a field. Additionally, some of the concerns around generality and dependence on the model’s underlying judgment capability, although partially addressed, need much deeper analysis in the main paper to complement the incremental efforts.

---

> > > ### Author Response · Authors · 2026-04-04
> > >
> > > We thank the reviewer for reading our response.
> > >
> > > We would like to clarify a few points and address the remaining concerns.
> > >
> > > > **On the characterization of our contribution as "leveraging pairwise comparison for evaluation and training."**
> > >
> > > Our paper does not claim pairwise comparison as novel. We explicitly discuss it as already established in Section 2 and Appendix A (related work). Our contribution is in studying and substantially improving *self-verification* capabilities, which is an important and largely unsolved problem. Recent work (Lu et al., 2025) shows that pointwise self-verification does not work, and the current literature does not contain techniques showing significant improvements to self-verification. We show that V1-Infer overcomes this limitation, and develop two new systems (V1-Infer, V1-PairRL) to make effective self-verification practical.
> > >
> > > While the pairwise primitive is known, our contributions lie in identifying why self-verification fails, designing novel algorithms that make it work (V1-Infer's Swiss tournament, V1-PairRL's co-evolving training with novel mechanisms to mitigate reward-hacking), and rigorously validating it across a broad set of settings (also see next point).
> > >
> > > > **On "reaching SOTA performance" not justifying a contribution.**
> > >
> > > We agree that SoTA alone is insufficient, and we did not frame our contribution this way. Our contributions are:
> > >
> > > - Identifying *why* self-verification fails (calibration collapse, diversity collapse)
> > > - Proposing an efficient inference algorithm that makes self-verification viable (V1-Infer)
> > > - Developing a co-evolving RL recipe to train a model as a generator and pairwise self-verifier, with novel mechanisms to mitigate reward hacking (V1-PairRL). No prior work shows a recipe for co-evolving training for inducing pairwise-self-verification capabilities
> > > - Validating this unified system across 4 models, 8+ benchmarks spanning code, math, and now, non-verifiable medical QA for both V1-Infer and V1-PairRL
> > >
> > > Performance improvements serve as evidence that our contributions are effective, not as the contribution itself. That said, developing a technique that leads to SoTA test-time scaling is extremely non-trivial.
> > >
> > > > **On "deeper analysis needed in the main paper."**
> > >
> > > We genuinely want to address this concern and would welcome specific guidance on what analysis would be helpful.
> > >
> > > Our rebuttal addressed all four original weaknesses with extensive new experiments: V1-PairRL on 3 domains including non-verifiable medical QA (W2), Math-RL and Rubric-RL baselines (W3), test-time scaling comparisons with 4 external verifiers and RSA (W3), showing that pairwise judgment capability is trainable with a +215% improvement on Math-RL (W4), and evaluation on harder and/or more ambiguous tasks including SWE-Bench-Lite and non-verifiable HealthBench-Hard (W4).
> > >
> > > These experiments required substantial compute (days of H100 node usage that we managed to get with extreme difficulty) which we invested because we took the concerns seriously.
> > >
> > > The ICML reviewer guidelines ([link](https://icml.cc/Conferences/2026/ReviewerInstructions)) encourage specific and actionable feedback. We would be grateful if the reviewer could point us to the particular gaps they see, so we can address them during the remaining discussion period.
> > >
> > > > **On novelty per ICML criteria.**
> > >
> > > ICML reviewer instructions state: "originality does not necessarily require introducing an entirely new method. Rather, a work that provides novel insights by evaluating existing methods, or demonstrates improved understanding is also equally valuable" and "Does this work offer a novel combination of existing techniques, and is the reasoning behind this combination well-articulated?"
> > >
> > > Our paper satisfies these criteria: we provide novel insights about self-verification failures, develop novel algorithms (V1-Infer's uncertainty-guided Swiss tournament, V1-PairRL to improve self-verification, along with mechanisms to mitigate reward-hacking to make this training practically possible), and articulate why V1-Infer and V1-PairRL succeed where pointwise fails, validated across 4 models and 8+ benchmarks with various degrees of hardness.
> > >
> > > We sincerely believe our rebuttal addressed all four weaknesses. The reviewer acknowledged ("results look solid and the engineering contributions are clear"), and we hope the clarifications above resolve the remaining concerns. We respectfully ask the reviewer to consider whether the concerns are sufficiently grounded to maintain a reject recommendation given the substantial new evidence.
> > >
> > > We thank the reviewer for their time, and remain happy to provide additional clarifications, experiments, or other specific results during the remaining discussion period.
> > >
> > > **Note: We would be happy to answer additional questions and provide experimental results and analysis. Please feel free to “edit” your response to the rebuttal above, and we will be happy to edit this response with the answers.**

---

### Decision · Program_Chairs · 2026-04-30

**Decision:**

Accept (regular)

**Comment:**

This paper proposes V1, a framework using pairwise self-verification and joint RL co-training to improve test-time scaling. Scores are 3, 4, 5, 4. Three reviewers recommend Accept, praising the strong empirical gains and elegant reward-hacking mitigations.
One reviewer maintained a Weak Reject (3), citing limited novelty of the pairwise primitive. I overrule this: adapting it to effectively solve self-verification failures is highly non-trivial. The extensive rebuttal further proved its robustness. I recommend acceptance.